# OTX2 represses sister cell fate choices in the developing retina to promote photoreceptor specification

**Miruna Georgiana Ghinia Tegla[1], Diego F Buenaventura[1,2], Diana Y Kim[1], Cassandra Thakurdin[1†], Kevin C Gonzalez[1], Mark M Emerson[1,2,3]\***

[1]Department of Biology, The City College of New York, City University of New York (CUNY), New York, United States; [2]PhD Program in Biology, The Graduate Center of the City University of New York (CUNY), New York, United States; [3]PhD Program in Biochemistry, The Graduate Center of the City University of New York (CUNY), New York, United States

**Abstract** During vertebrate retinal development, subsets of progenitor cells generate progeny in a non-stochastic manner, suggesting that these decisions are tightly regulated. However, the gene-regulatory network components that are functionally important in these progenitor cells are largely unknown. Here we identify a functional role for the OTX2 transcription factor in this process. CRISPR/Cas9 gene editing was used to produce somatic mutations of OTX2 in the chick retina and identified similar phenotypes to those observed in human patients. Single cell RNA sequencing was used to determine the functional consequences OTX2 gene editing on the population of cells derived from OTX2-expressing retinal progenitor cells. This confirmed that OTX2 is required for the generation of photoreceptors, but also for repression of specific retinal fates and alternative gene regulatory networks. These include specific subtypes of retinal ganglion and horizontal cells, suggesting that in this context, OTX2 functions to repress sister cell fate choices.

**\*For correspondence:**
memerson@ccny.cuny.edu

**Present address:** †Doctoral Program in Neurobiology and Behavior, Columbia University, New York, United States

**Competing interests:** The authors declare that no competing interests exist.

## Introduction

The vertebrate retina is a highly structured tissue comprising six neuronal classes and several types of glia. The specification of these diverse cell types is a dynamic process, largely conserved across vertebrate species, with the commitment to a final fate of retinal ganglion cells (RGCs), cone photo-receptors (PRs) and horizontal cells (HCs) in the first developmental wave, followed progressively by amacrine cells (ACs), rod PRs and bipolar cells (BCs) (*Masland, 2001*; *Bassett and Wallace, 2012*). Müller glia are also generated during the last phase of retinal development, while astrocytes and the resident microglia are generated at early stages, independent of the neuronal retina, simultaneously with optic stalk formation (*Kolb, 2007*; *Tao and Zhang, 2014*).

Elucidation of the mechanisms by which each cell type is generated from multipotent retinal pro-genitor cells (RPCs) is crucial for the development of successful cell replacement therapies for patients that suffer from retinal degeneration. Initial studies using viral labeling showed that multipo-tent RPCs have the potential to give rise to all retinal cell types, in a mechanism that can be explained by both deterministic and probabilistic models (*Turner and Cepko, 1987*; *Wetts and Fraser, 1988*; *Holt et al., 1988*; *Cepko et al., 1996*; *Vitorino et al., 2009*; *He et al., 2012*). Recently, identification of markers of restricted, neurogenic lineages enabled the characterization of these types of RPCs. For example, one class of restricted RPCs gives rise to all cell types except RGCs (*Brzezinski et al., 2011*), one produces all cell types except RGCs and Müller glia (*Hafler et al., 2012*), and another that is largely restricted to only two cell types – cone PRs and HCs

(*Hafler et al., 2012*; *Emerson et al., 2013*; *Suzuki et al., 2013*; *Schick et al., 2019*). This last RPC type is defined by the activity of the cis regulatory module 1 (CRM1) of the gene THRB. Previous studies have determined that the activation of the THRBCRM1 element requires the expression and direct binding of the Orthodenticle Homeobox 2 (OTX2) and ONECUT1 transcription factors (*Emerson et al., 2013*; *Souferi and Emerson, 2019*). Moreover, a recent study has identified cellular and molecular changes that occur during the generation of these fate-restricted RPCs, but not in multipotent RPCs (*Buenaventura et al., 2018*). However, the roles of OTX2 and ONECUT1 in the establishment of this restricted signature are not well understood.

OTX2 is one of the key regulators of nervous system development, as it is involved in forebrain and midbrain specification (*Acampora et al., 1995*; *Ang et al., 1996*; *Simeone et al., 2002*), pineal and pituitary gland development (*Nishida et al., 2003*; *Henderson et al., 2009*), as well as development of sensory structures such as the inner ear (*Matsuo et al., 1995*) and visual system (reviewed in *Beby and Lamonerie, 2013*). In the visual system, OTX2 is expressed in different subsets of cells as development progresses – first, during optic vesicle formation, OTX2 directs evagination of the optic vesicle to contact the surface ectoderm along with RAX, PAX6, HES1 and SIX3 (*Adler and Canto-Soler, 2007*). Later, during the specification of the neuronal and retinal pigment epithelium (RPE) territories, OTX2 is highly expressed in the RPE (*Martinez-Morales et al., 2001*) and in a subset of RPCs that primarily generate cones and HCs (*Emerson and Cepko, 2011*; *Emerson et al., 2013*; *Buenaventura et al., 2018*), then in postmitotic PRs during the specification of retinal neuronal cell types (*Nishida et al., 2003*), in the PRs and BCs in the mature retina (*Koike et al., 2007*; *Kim et al., 2008*), and in a subset of Müller glia (*Brzezinski et al., 2010*). As a consequence of its involvement during early eye genesis, several studies have reported OTX2 mutations in humans with ocular malformations. The clinical manifestations range from unilateral and bilateral anophthalmia, microphthalmia, optic nerve aplasia to various forms of coloboma (reviewed in *Gat-Yablonski, 2011*).

Previous studies have examined the role of OTX2 in the development of the neuronal retina using conditional floxed mouse models, as homozygous OTX2 mutants die embryonically due to defects in the specification of the anterior neuroectoderm (*Acampora et al., 1995*). When OTX2 loss is mediated by a post-mitotic PR Cre driver, a severe loss of PRs was observed, while the number of PAX6-positive cells was increased, suggesting a trans-differentiation of the mutant cells into amacrine-like neurons (*Nishida et al., 2003*). Microarray analysis of these retinas supported this conclusion (*Omori et al., 2011*). Similar PR loss and abnormal generation of amacrine-like cells, along with loss of BCs and HCs was reported, when OTX2 ablation was initiated throughout the early retina (*Sato et al., 2007*). Despite the contributions from these studies that implicate OTX2 as a crucial regulator of PR development, several questions remain unanswered. Specifically, what are the cell fates and underlying gene regulatory networks that are impacted by loss of OTX2? While previous analyses have suggested that ACs are ectopically induced upon loss of OTX2, these effects were characterized in the postnatal retina leaving the primary effects of OTX2 loss unknown. In addition, OTX2 expression is initiated in specific RPC populations prior to the formation of PRs and this expression would not be compromised by the previously used PR Cre drivers (*Emerson et al., 2013*; *Buenaventura et al., 2018*). Thus, the functional role of OTX2 in this RPC population that will generate PRs is unknown.

This study reports a characterization of OTX2 mutants during various stages of retinal development with a focus on OTX2-positive RPCs. CRISPR-induced OTX2 loss-of-function mutations were introduced in the context of the developing chicken retina in vivo and ex vivo, providing a robust and facile experimental system to determine OTX2 function. Successful ablation of OTX2 was demonstrated by immunofluorescence, gene expression analysis and deep sequencing of the OTX2 gene. The effects of OTX2 loss on OTX2-positive RPCs were analyzed using a reporter active in this cell population combined with the examination of known markers and characterization using single cell RNA sequencing. In addition, these data show the strict requirement of OTX2 for the specification of PR cells, while specific subtypes of RGCs and HCs are generated in the OTX2 mutant retinas, suggesting that alternate fates are not randomly chosen, but are selected from those normally associated with OTX2-positive RPCs. This high-resolution analysis of OTX2 function provides new insights into the process of PR formation in the vertebrate retina and the mechanisms of retinal cell fate choice.

## Results

### CRISPR/Cas9-induced ablation of the OTX2 gene at the optic vesicle stage results in anatomical defects of the overall eye structure, pigment epithelium and neuronal retina

To generate mutations that disrupt the function of the OTX2 gene, two RNA guides that corresponded to mRNA positions 141 and 167 in the coding sequence of the OTX2 gene, called guide 2 (g2) and guide 3 (g3), were used (*Figure 1A–B*). In all experiments, a DNA plasmid (plasmid p18) encoding one of these guides and an encoded source of Cas9 was co-electroporated with a fluorescent reporter into several stages of chick retinas, either in ovo or ex vivo, and analyzed at specific stages (*Figure 1C–E*). Throughout the study, the comparison was done between retinas electroporated with the CRISPR guide plus electroporation control vector mix and retinas electroporated with the same electroporation control as well as an 'empty' p18 plasmid that contained the Cas9 open reading frame driven by the CAG promoter and the guide RNA scaffold, but without the 20 bp part of the guide RNA specific to the target gDNA. For simplicity, the control eyes will be referred to as CTRL, while the experimental ones as OTX2^CRISPR.

In ovo electroporation was first used at E1.5 (HH9-HH11), targeting only the right eye of each embryo. Electroporation of the OTX2 g2 or g3 yielded severe anatomical defects of the electroporated eye upon analysis at E5, a timepoint when both the neuronal retina and the RPE territories have already been specified (*Figure 1C*, *Figure 1—figure supplement 1A–H*). The morphological defects observed in the OTX2^CRISPR mutants ranged from microphthalmia (*Figure 1—figure supplement 1C,E,I*), patches of depigmentation in the RPE (*Figure 1—figure supplement 1C–H*), defects of the optic nerve (ON), such as ON head enlargement, as well as both iridial and optic stalk coloboma-like phenotypes (*Figure 1—figure supplement 1D,E,F,H*). For all in ovo experiments the left eye, not targeted by electroporation, developed normally as expected (MGT). Quantification of the above-mentioned phenotypes is shown in *Figure 1—figure supplement 1I–J* and *Supplementary file 2*. Similar results were obtained in seven independent experiments.

The mosaicism observed in the patches of depigmentation in the mutant RPE as well as the variable size of the mutant eyes reflects the electroporation efficiency as well as the fact that the mutations were induced through non-homologous end-joining (NHEJ) after the Cas9 induced double-stranded break, and, therefore, the clones derived from electroporated cells could be different with respect to their targeted mutation.

To assess the structural defects observed in the OTX2^CRISPR mutants, an immunofluorescence-based characterization was performed on vertical sections of retinas electroporated at E1.5 and analyzed at E5. Mutants varied in thickness of both RPE and neuronal retina (*Figure 1F–I* and *Figure 1—figure supplement 1K–P*). The RPE layer was found to contain structures composed of cells that were depigmented, enlarged, intensely GFP-positive, and OTX2-negative (*Figure 1F,G*). These structures were absent in CTRL retinas and were largely EdU and PH3-negative, with low expression levels of VSX2 and high levels of PAX6 (*Figure 1H,I*, *Figure 1—figure supplement 1K–P*).

At the developmental time of analysis, OTX2-positive cells in the neuronal retina are normally observed in a dispersed pattern in the outermost part of the retina (*Buenaventura et al., 2018*; *Figure 1F*). The OTX2^CRISPR mutants displayed varying degrees of effects on OTX2 expression, ranging from absence of the outer OTX2 population to complete loss of OTX2 throughout the entire retina (*Figure 1G*).

As noted, the mutations induced during optic cup formation yielded a broad range of morphological defects due to the mutation of OTX2 in both neuronal retina and RPE, and therefore, obfuscated the functional evaluation of OTX2 mutation during PR development. To more specifically target the neuronal retina, in ovo electroporation experiments were performed at E3 (HH18), when the RPE and neuronal territories are already specified and the majority of cells are RPCs with high mitotic potential, ensuring the propagation of the mutations to many daughter cells.

The CAG::GFP plasmid was used as an electroporation control to identify the targeted cells. At this stage, the electroporated area is limited due to the nature of the experiment (Methods), and the sparse clones of electroporated cells can therefore be analyzed in the context of a mostly unaffected retina (*Figure 1J*). When mutant retinas were analyzed at E6, the analyzed regions of the retina have many OTX2-positive cells, however, the GFP-positive cells generally lack OTX2 protein (*Figure 1K*).

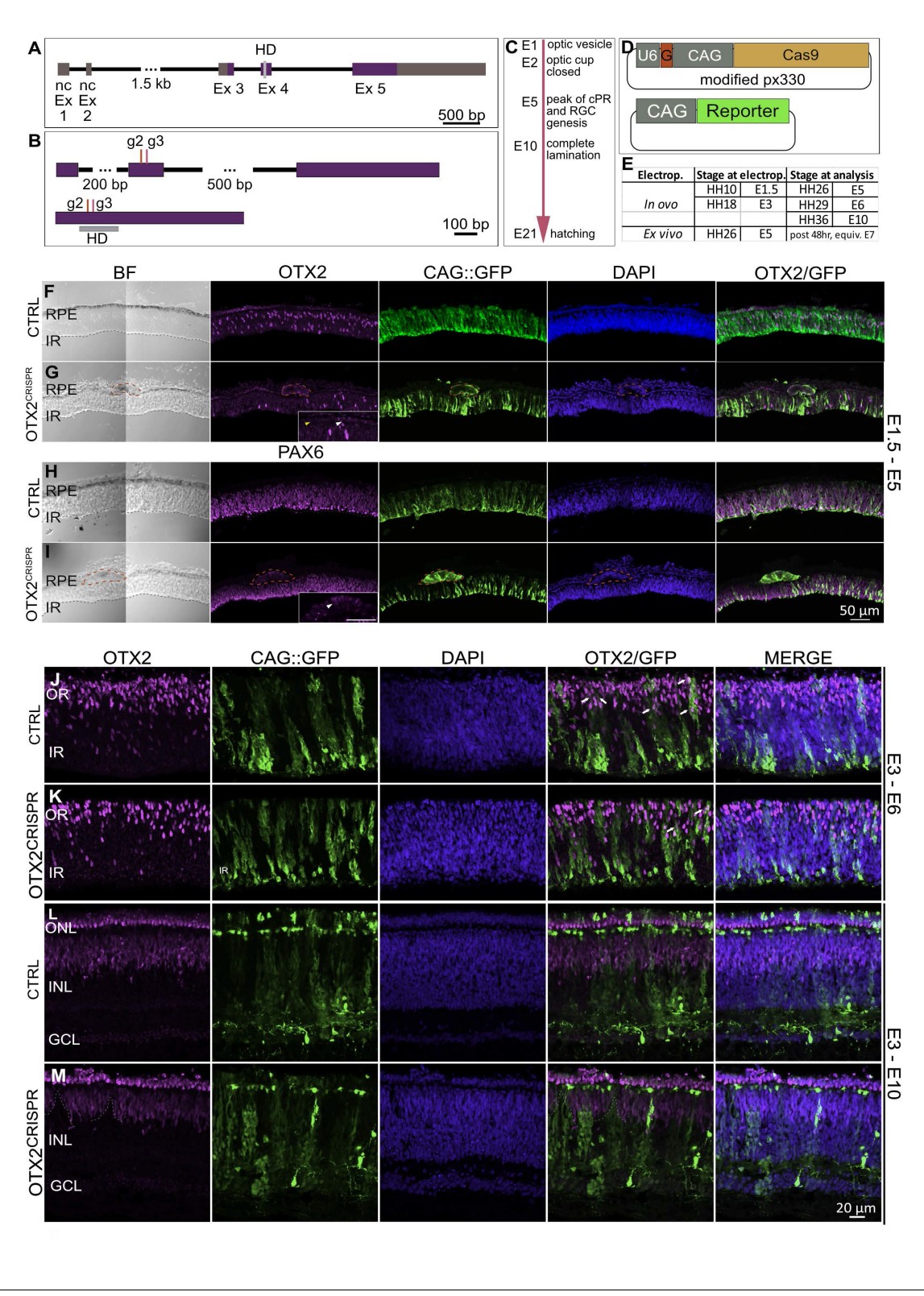

**Figure 1.** OTX2[CRISPR] guide design and targeted electroporation of the early chick eye yields severe abnormalities of the retina and RPE. (**A**) Schematic representation of the *Gallus gallus* OTX2 genomic locus. Purple blocks represent coding exon regions. Gray blocks represent non-coding exon regions. Light grey bar in exon 4 represents homeodomain region. (**B**) Location of guides 2 and 3 relative to the unspliced (top) and spliced (bottom) OTX2 mRNA. Grey box shows the mRNA regions that encode the homeobox domain. (**C**) Key events in the developmental timeline of the eye development

*Figure 1 continued on next page*

Figure 1 continued
in chick. (D) Schematic of co-electroporated plasmids. U6 is the promoter for the guide RNA, denoted by G., CAG drives expression of Cas9 and fluorescent reporters. (E). Time points for electroporation of CRISPR plasmids and analysis. (F–I) Confocal microscopy analysis of CTRL and OTX2[CRISPR] g2-induced mutant retinal sections targeted at E1.5 and analyzed at E5. OTX2 protein expression in CTRL (F) as compared to Mutant (G). Mutant RPE is depigmented and cells with strong GFP and low levels of OTX2 are identified by red outline. White arrow in high magnification insert shows OTX2-positive cells, whereas the yellow arrow point to cells that are negative for OTX2. (H, I) CTRL (H) and Mutant (I) sections stained for PAX6. RPE structures in mutants are outlined by dotted lines and shown as a high magnification insert in (I). (J–M) Qualitative analysis of CTRL and g2 retinas electroporated in ovo at E3 and analyzed at E6 (J–K) and E10 (L–M). (J–K) White arrows denote examples of electroporated cells that are positive for OTX2. (L–M) GFP-positive, OTX2-negative patches (dotted lines) are present in the INL and PR layers of OTX2[CRISPR] mutants. Ex, Exon; nc, non-coding; HD, homeodomain; BF, brightfield; RPE, retinal pigment epithelium; IR, inner retina, OR, outer retina, ONL, outer nuclear layer, INL, inner nuclear layer, GCL, ganglion cell layer.

The online version of this article includes the following figure supplement(s) for figure 1:

**Figure supplement 1.** Effects of OTX2[CRISPR] mutation induced at the optic vesicle stage.

The absence of OTX2-positive cells in the mutant was more pronounced at E10, when the retina is layered, and the OTX2-positive PRs are concentrated in the ONL and BCs are located in the outer part on the INL (*Figure 1L*). Columns of GFP-positive cells that lack OTX2 form gaps within the OTX2-positive populations in the ONL and INL (*Figure 1M*). These experiments suggest that OTX2 g2 guided-Cas9 is able to disrupt OTX2 expression and leads to underrepresentation of PR and BC fates in the electroporated population, in agreement with a previous study in mice (*Nishida et al., 2003*; *Koike et al., 2007*). To quantify these effects and investigate the fate of OTX2 mutant cells, an ex vivo preparation was used at a later timepoint.

## Highly effective OTX2 mutagenesis leads to a reduction in photoreceptors and increase in PAX6 positive cells

To more precisely examine the role of OTX2 during PR genesis, the OTX2 CRISPR g2/Cas9 and CAG::GFP plasmids were electroporated into embryonic day 5 (E5) chicken retinas. In parallel, several experiments were replicated using g3 to validate observed phenotypes (*Figure 2—figure supplement 1*). Retinas were electroporated ex vivo and cultured for two days. Immunofluorescence examination of retinas revealed a qualitative decrease in OTX2-positive cells in the electroporated population (arrows in *Figure 2A–B*).

To quantify the effects presented in *Figure 2A–B*, similar experiments were performed, followed by dissociation of the entire retina and quantification by flow cytometry of the OTX2-positive cells detected with antibodies against OTX2. In the OTX2[CRISPR] mutants, an approximately 50% reduction in the percentage of OTX2-positive cells within the electroporated population was detected using two different OTX2 antibodies (*Figure 2C–D*, *Supplementary file 2*, *Figure 2—figure supplement 1A,B*).

In addition, the percentage of PAX6-positive cells in the OTX2[CRISPR] mutants was investigated, since previous experiments that used a mouse model of OTX2 mutation identified an upregulation of this gene (*Nishida et al., 2003*). In OTX2[CRISPR] mutants, an almost two-fold increase in the percentage of PAX6-positive cells was detected as compared to controls (*Figure 2E–F*, *Supplementary file 2*).

To quantify the effect on PRs from OTX2 mutagenesis, the number of cells that activate the THRBCRM2 cis-regulatory element was analyzed. This element is associated with the THRB gene and is active in a subset of cone PRs in the chick retina (*Emerson et al., 2013*; *Schick et al., 2019*). OTX2[CRISPR] mutant retinas had a significant reduction in THRBCRM2 activity (*Figure 2G–H*, *Figure 2—figure supplement 1C,D*). Thus, this assessment suggests that the OTX2[CRISPR] mutation leads to a robust reduction of OTX2 protein as well as a PR reporter. Quantification of all parameters reported above are detailed in *Supplementary file 2*.

## The OTX2ECR2 reporter reveals a change in cell fate in the OTX2[CRISPR] cells

As OTX2 expression has been reported to be initiated in RPCs that predominantly generate cone PR and HCs, the fate of the daughter cells of these RPCs was examined upon OTX2 mutation. To do this, the previously reported OTX2ECR2 enhancer element (*Figure 3A*), which is active in OTX2-

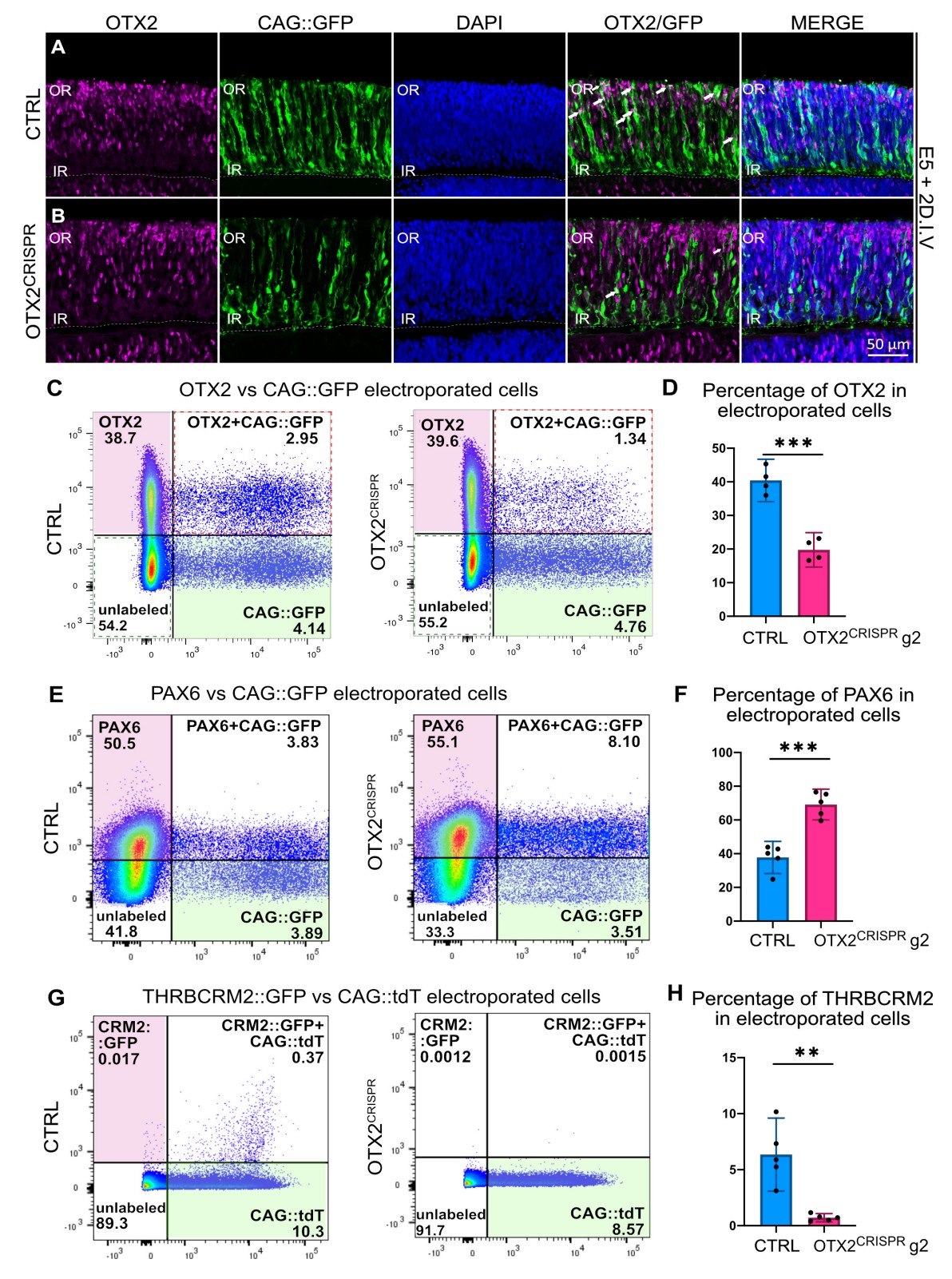

**Figure 2.** Highly effective CRISPR-induced mutation of OTX2 in E5 retinas yields severe reduction in photoreceptor markers and increase in the number of PAX6-positive cells. (**A–B**) Confocal microscopy assessment of vertical sections of retinas co-electroporated with CAG::GFP and CRISPR plasmid, and then immunostained for OTX2 (magenta), GFP (green) and DAPI (blue) (**A**) Control (**B**) OTX2$^{CRISPR}$ g2. White arrows denote electroporated OTX2-positive cells. (**C–H**) Representative dot plots showing the overlap between OTX2 (**C**), PAX6 (**E**), and THRBCRM2::GFP (**G**) with CAG::GFP or CAG::tdT

*Figure 2 continued on next page*

*Figure 2 continued*

in CTRL and OTX2^CRISPR g2 dissociated retinas. (D, F, H) Bar graph showing the average percentage of cells positive for each marker out of the total number of electroporated cells detected with GFP or tdT. Each point represents one biological replicate (n ≥ 4 for both CTRL and OTX2^CRISPR g2 and error bars represent 95% confidence intervals. *** represents p<0.0001 and ** represents p<0.001; OR, outer retina; IR, inner retina. DIV, days in vitro. The online version of this article includes the following figure supplement(s) for figure 2:

**Figure supplement 1.** Flow cytometry quantification of OTX2 immunoreactivity and THRBCRM2 activity in OTX2^CRISPR g3 retinas.

positive RPCs in the chicken retina, was used (*Emerson and Cepko, 2011*). In CTRL retinas, upon electroporation at E5 and two days in culture, GFP expression driven by the enhancer's activity is observed predominately in the outer retina/PR layer, with fewer cells, showing less intense GFP expression, in the inner retina (*Figure 3B*). The enhancer's activity, observed by GFP expression, co-localizes primarily with OTX2-positive retinal cells electroporated at E5 and cultured for 2 days, consistent with previous observations (*Emerson and Cepko, 2011*). In vivo experiments have demonstrated that OTX2ECR2 drove reporter activity in both PRs and LHX1-positive HCs and very rarely targeted POU4F1-positive RGCs (*Emerson and Cepko, 2011*).

Characterization of OTX2ECR2 reporter activity in the OTX2^CRISPR retinas shows a qualitative shift in the cell populations – a dramatic reduction in the PR population as well as an increase in the inner retina population with many more intensely labeled GFP-positive cells (*Figure 3C–D*).

To confirm previously reported estimates of the activity of the OTX2ECR2 element in OTX2-positive cells, a quantitative flow cytometry analysis was performed. 73% (c.i. ±2.4%) of GFP-positive cells were also OTX2-positive. The remaining 27% cells likely represent cells with either nonspecific activation of the element, or cells in which the reporter was previously active. In contrast to CTRL samples, OTX2ECR2::GFP-positive cells in OTX2^CRISPR mutants were positive for OTX2 protein in only 22% (c.i. ±5.9%) of the g2 cells and in 15% (c.i. ±8.1) of g3 cells (*Figure 3E–G*, *Supplementary file 2*). In addition, there is a robust increase in both the number and the intensity of the signal in the GFP-only population (green box), suggesting that these are the newly generated, mutant cells (*Figure 3E*).

## Single cell RNA sequencing analysis of the OTX2ECR2-positive cells reveals widespread changes in the distribution of cells per cluster

To further investigate the role of OTX2 in restricted RPCs, a single cell transcriptome analysis was performed on the OTX2ECR2::GFP-positive population from OTX2^CRISPR g2 mutant and CTRL retinas (*Figure 5—figure supplement 1A,B*) after ex vivo electroporation at E5 and two days in culture. Here also, the CTRL retinas were electroporated with the empty p18 vector, lacking the guide sequence, but containing the scaffold RNA, as well as Cas9 coding region, driven by the CAG promoter. The use of the OTX2ECR2 reporter allowed an in-depth analysis of this selected population, enriching for PRs and restricted RPCs.

Cell suspensions of dissociated retinas were processed for single cell library preparation using a 10X Genomics Chromium Single Cell 3' platform (*Zheng et al., 2017*) and sequenced on an Illumina HiSeq 4000 sequencer. Prior to retinal dissociation, the embryos were genotyped for sex determination (Methods) and cells from three retinas of embryos PCR-identified as female were pooled per sample. For each of the mutant retinas included in the sample, the contralateral eye was used as a CTRL sample to limit the variability across individual animals.

To estimate the extent of OTX2 mutagenesis, the sequencing reads around the Cas9 target site were analyzed in both the mutant and CTRL samples (*Figure 5—figure supplement 1C,D*). Examination of the sequencing reads using the Integrative Genomics Viewer - IGV (*Robinson et al., 2011*) shows a wide degree of mutation present at that location in the OTX2^CRISPR sample that is consistent with repair through the NHEJ pathway (*Figure 5—figure supplement 1E*). However, the majority of reads are located at the 3' end of the gene due to the particular library preparation method used (*Figure 5—figure supplement 1C*).

To further quantify these mutations, OTX2 mRNA was isolated from the same samples used for single cell gene expression profiling. Amplicons that contained the region targeted by guide 2/Cas9 were made from cDNA. Deep sequencing showed that the majority of reads in the OTX2^CRISPR cells contained a mutation localized to the targeted region with only 28.69% of reads containing a

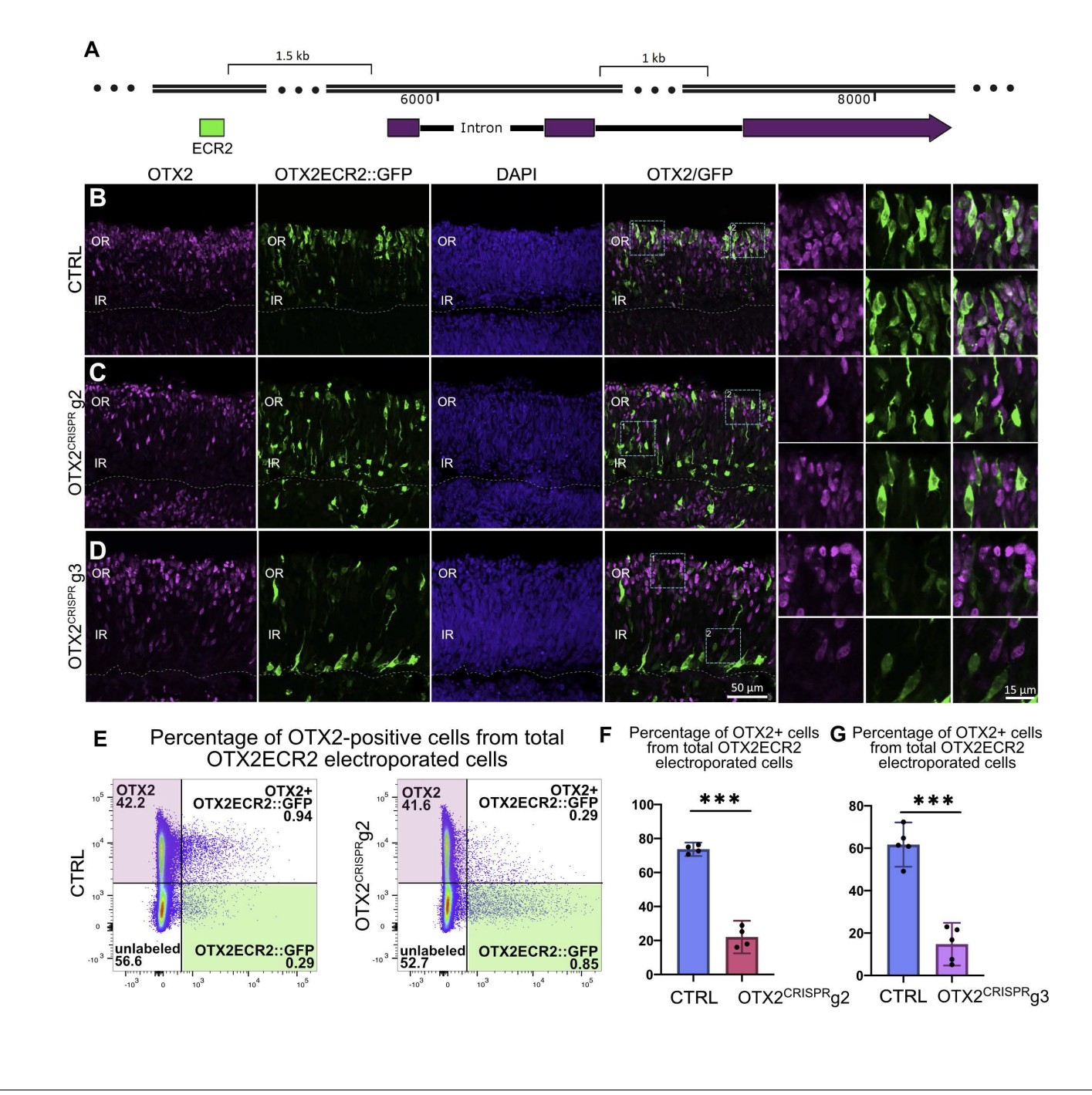

**Figure 3.** OTX2ECR2 reporter expression in CTRL and OTX2^CRISPR retinas. Retinas electroporated at E5 with OTX2ECR2::GFP and control or OTX2^CRISPR plasmids and analyzed after 48 hr. (**A**) Schematic representation of the chick OTX2 genomic locus, and the location of the ECR2 element, 1.5 kb upstream of the start codon. (**B–D**) OTX2ECR2::GFP reporter predominantly labels cells in the outer CTRL retina (**B**). This population is markedly reduced in both OTX2^CRISPR mutants (g2 in (**C**) and g3 in (**D**)), and a new population located in the inner retina is formed. High magnification views of the regions boxed in the OTX2/GFP panels are shown in the rightmost panels. (**E**) Representative dot plots showing the overlap between OTX2ECR2:: GFP (x-axis) and OTX2 protein (y-axis). The population of OTX2ECR2::GFP-positive cells that express OTX2 protein decreases in OTX2^CRISPR g2 retinas. (**F**) Quantification of the OTX2+/GFP+ cells from the total GFP+ cells for OTX2^CRISPR g2 (**F**) and g3 (**G**) compared to the controls. Error bars represent 95% confident intervals. *** represents p<0.0001, n ≥ 4 for both comparisons and each point represent one biological replicate. OR, outer retina; IR, inner retina.

wildtype sequence (*Figure 5—figure supplement 1F*). In contrast, the CTRL samples contained 97.97% wildtype sequence.

The amplicon sequencing recovered the following distribution of mutations: 25.11% (Reads 2, 7, 12) are deletions without downstream frameshift effects, 21.46% (Reads 4, 5, 6, 10, 11) are deletions with downstream frameshift effects, and 21.11% (Reads 3, 8, 9) are insertions with downstream frameshift effects (*Figure 5—figure supplement 1H*). From the total number of reads 3.62% were noisy, and therefore were excluded from the quantification. Therefore, more than 40% of the mutations are expected to lead to truncation of the protein within the homeodomain and the remainder to alter the sequence in the homeodomain.

Using the single cell RNA sequencing analysis, ten different clusters were identified in the CTRL OTX2ECR2 lineage based on previously characterized markers (*Figure 4A*). As expected, not all clusters presented high levels of OTX2, supporting the previous data on the characterization of the OTX2ECR2 enhancer (*Emerson and Cepko, 2011*). Clusters of restricted RPCs (restrRPC1 and restrRPC2), conePR1, conePR2, and cone homotypical PC (chPC) show high levels of OTX2 in the majority of cells, whereas clusters multiRPC (multipotent RPC), HC1, HC2, AC/HC/RGC and RGC show only sparse cells positive for the marker, with minimal expression levels (*Figure 4A–B*). These could represent cells that either have a previous history of OTX2 expression, an inappropriate activation of the OTX2ECR2 element, or cells that were inadvertently collected during flow cytometry.

Two OTX2 and OLIG2 positive clusters were identified as containing restricted RPCs - restrRPC1 with cells undergoing G2M or S-phase, (21% from total), and rest RPC2 with cells in G1 phase (12.04%), although very finely delimited in terms of the markers they express (*Figure 4C–E*). Based on gene expression profiles, these are likely to represent the same cell state in different stages of the cell cycle. 6.9% from total were assigned to be multipotent retinal progenitor cells (multi RPC) based on their high expression of VSX2, NR2E1, ID1 and SOX9.

Two different clusters – cone PR1 (12.36%) and cone PR2 (15.58%) are assigned to be PRs, based on the markers expressed. The chPC cluster was defined as cone homotypical progenitor cells (chPC) (14.74%) based on their very similar gene expression profiles to the other PR clusters, but identified by marker expression to be in G2/M and S-phase. All three clusters present high levels of RBP4, with expression of THRB, RXRG, and ISL2 in clusters cone PR2 and chPC and lower levels of RXRG in cone PR1 (*Figure 4D*).

Cluster RGC was identified as containing RGCs (3.54%), based on its high expression of POU4F2 and POU4F3, RBPMS2, as well as members of the EBF family. Surprisingly, POU4F1, which is present in a large fraction of RGCs was only found in a minority of the cells in this cluster (*Liu et al., 2000*). At least two clusters were identified to be HCs, cluster HC1 and HC2 (5.61% and 4.56%, respectively), both containing cells that express PROX1, ONECUT1, TFAP2A and TFAP2B. The 2.76% in cluster AC/HC/RGC express the horizontal/amacrine cell (AC) marker PTF1A almost exclusively, but also members of the DLX family of transcription factors, known to be characteristic for murine ACs and RGCs (*de Melo et al., 2003*). However, due to the similarity in their transcription programs and to the lack of specific markers for ACs at this developmental time point, the discrimination between the three classes is difficult to make (*Figure 4E*, *Figure 5—figure supplement 2*, *Supplementary file 4*; *Clark et al., 2019*).

To compare the single cell transcriptomes from both CTRL and OTX2^CRISPR samples, a similar cluster analysis based on the principal component was run with the two datasets combined (*Figure 5A*). The restricted RPCs were classified in one cluster, as expected from the minimal differences observed when the CTRL dataset was analyzed alone. In addition, a new cluster was defined, containing cells that were only found in the mutant retinas.

OTX2 mRNA expression was reduced in the OTX2^CRISPR sample but not completely lost in the mutant cells (*Figure 5D*). The cells that continue to express OTX2 mRNA have a variety of mutations captured in reads two to twelve (R2-R12) (*Figure 5—figure supplement 1H*). These mutations might lead to the loss of functional OTX2 protein and generate protein with various defective functions.

Widespread changes in the distribution of cells per clusters of the OTX2^CRISPR sample relative to the CTRL clusters were observed (*Figure 5A–C,E,F. Figure 6—figure supplement 1B–C*). The restricted RPC populations were not severely changed, either in terms of percentages of cells (29% in CTRL and 25% in the OTX2^CRISPR dataset), or in regards to their cell cycle states (*Figure 5C*, *Figure 6—figure supplement 1C*). The multipotent RPC cluster was also unaffected in terms of number

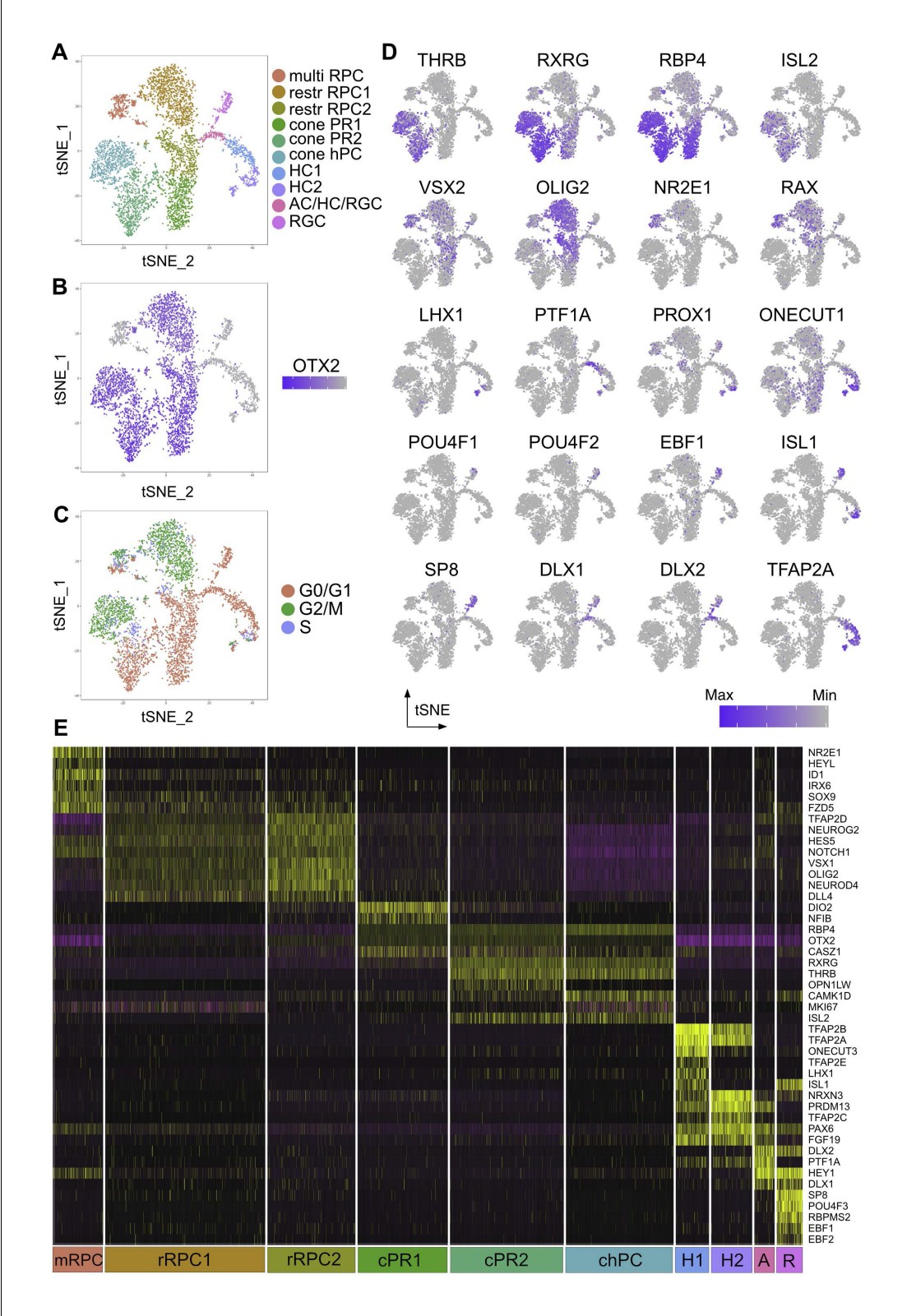

**Figure 4.** Single cell analysis of CTRL OTX2ECR2::GFP cells. (**A**) TSNE plots of the 10 clusters generated by the unsupervised algorithm Seurat, based on the gene expression of each cell analyzed. (**B**) Heatmap of OTX2 expression (purple) in the cells analyzed. (**C**) TSNE plots of the clusters showing their cell cycle state, G1, G2M or S-phase. (**D**) Heatmaps showing the expression of different markers in the 10 clusters, in TSNE view. (**E**) Heatmap of

*Figure 4 continued on next page*

*Figure 4 continued*

differentially expressed genes across the 10 clusters, where purple represents low gene expression and yellow represents high expression. MultiRPC – multipotent RPC, restrRPC – restricted RPC.

of cells (7% cells in CTRL, 6% in the OTX2$^{CRISPR}$) and in terms of gene expression (*Figure 5F*, *Figure 5—figure supplement 2*, *Supplementary file 5*).

## OTX2 mutation leads to an increase in specific subtypes of horizontal and retinal ganglion cells and loss of photoreceptors

All three PR clusters were massively reduced in the OTX2$^{CRISPR}$ sample compared to the CTRL one – from 18% to 1% in cluster cone PR1, from 14% to 1% in cluster cone PR2 and from 16% to 1% for the cone homotypic PCs (*Figure 5C,F*). The majority of PR markers such as THRB, RXRG, RBP4, ISL2, OTX5 were markedly decreased (*Figure 5E*, *Figure 6A*, *Figure 5—figure supplement 2*). This finding supports the data from previously published mouse models of conditional OTX2 knockout (*Nishida et al., 2003*; *Sato et al., 2007*). However, there were significant changes in the representation of both HCs and RGCs that were not previously identified. The combined analysis revealed that the two HC clusters also changed in specific ways. Strikingly, the LHX1-positive HCs – HC1, increased approximately by 5-fold in the OTX2$^{CRISPR}$ sample, from 2% to 11%, whereas the percentage of cells in the ISL1-positive cluster HC2 decreased from 8% in the CTRL to 4% in the mutants (*Figure 5C,F*, *Figure 6B*). In addition, cluster AC/HC/RGC increased in the OTX2 mutants, from 3% to 5% mutant dataset (*Figure 5C,F*). This increase also suggests that the cluster is formed predominately by early RGCs and LHX1-positive HCs.

The number of RGCs in the OTX2$^{CRISPR}$ sample increased by approximately 4-fold, from 3% in the CTRL retinas to 12% in the mutants (*Figure 5F*, *Figure 5—figure supplement 2*). Surprisingly, there was no change in the number of POU4F1 positive RGCs, while the number of POU4F2 and POU4F3 positive cells accounted for all of the RGC increase in the OTX2$^{CRISPR}$ cells (*Figure 6C*).

To validate the results obtained from the transcriptomic analysis, a quantitative analysis was done on several markers – RXRG (labeling cone PRs), POU4F1 (labeling a sub population of RGCs), pan-POU4F (labeling POU4F1, POU4F2 and POU4F3 RGCs) and LIM1 (labeling LHX1-positive HCs). OTX2$^{CRISPR}$ complexes that contained either g2 or g3 were electroporated at E5 along with the OTX2ECR2::GFP reporter. Empty p18 plasmid was electroporated in the case of the control retinas.

Quantification of RXRG expression showed a reduction from 49.69% (c.i. ±6.9) RXRG-positive OTX2ECR2 positive cells in the CTRL cells to 20.74% (c.i. ±4.29) in the OTX2$^{CRISPR}$ g2 retinas, and 17.78% in the OTX2$^{CRISPR}$ g3 (c.i. ±3.73) retinas respectively, p<0.001 (*Figure 6D–G*; *Supplementary file 2*). A LIM1 antibody was used to determine the number of OTX2ECR2 LHX1-positive HCs. Quantification confirmed a significant increase in this population, from approximately 4.85% (c.i. ±0.48) in the CTRL retinas to 14.61% (c.i. ±2.2) in OTX2$^{CRISPR}$ g2 retinas, and 9.79% (c.i. ±1.83) in OTX2$^{CRISPR}$ g3 retinas, p=0.0003 for g2 and p=0.0014 for g3 (*Figure 6H–K*). The POU4F1 cells were quantified using a POU4F1 antibody. In agreement with the single cell analysis, the number of OTX2ECR2 POU4F1-double positive cells did not vary significantly when the two mutants were compared to the CTRL (*Figure 6—figure supplement 1D–G*; *Supplementary file 2*). The increase in numbers of POU4F2 and POU4F3-positive cells was validated using a panPOU4F antibody. The marker was expressed in 16% (c.i. ±0.46) OTX2ECR2-positive cells in the CTRL retinas, in 49.9% (c.i. ±04.81) in OTX2$^{CRISPR}$ g2 and in 32.4% (c.i. ±9.59) in OTX2$^{CRISPR}$ g3 retinas, respectively, p<0.001 for g2 and p=0.029 for g3 (*Figure 6 L-O*).

Taken together, these data confirm the increase in the number of specific cell fates, including LHX1-positive HCs and POU4F2/POU4F3 RGCs, while other populations such as multipotent RPCs, ISL1-positive HCs, and POU4F1 RGCs are relatively unchanged.

## Upregulation of specific genes in response to OTX2 CRISPR/Cas9 targeting

While the above analysis identified OTX2$^{CRISPR}$-induced changes in the number of cells that corresponded to particular cell fates, transcriptional changes that appeared to be unlinked to cell fate were also noted. One example was for the restricted RPC population, where unsupervised clustering

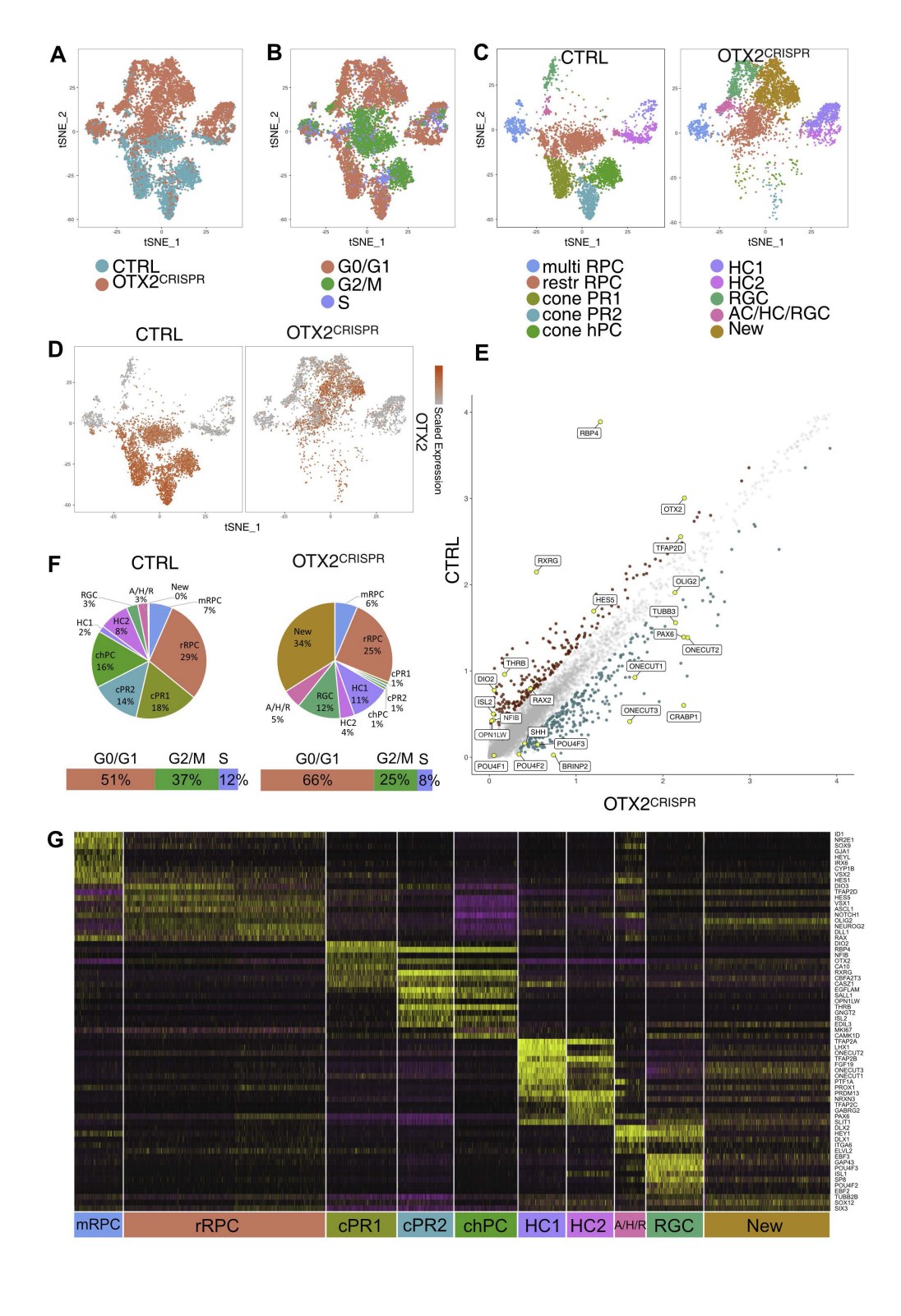

**Figure 5.** Combined single cell analysis from CTRL and OTX2^CRISPR retinas labeled by OTX2ECR2::GFP reporter. TSNE plots of the two datasets analyzed simultaneously (A) and labeled for their cell cycle signature (B). (C) TSNE plots of the 10 clusters determined by Seurat - multipotent RPCs (multiRPC), restricted RPCs (restrRPC), cone PR1 and 2, cone homotypical PC, HC clusters (HC1 and HC2), RGC, AC/HC/RGC and New, in the CTRL (left) and mutant (right) samples. (D) Heatmap of OTX2 expression across the two datasets. (E) Differentially expressed genes in the CTRL and

*Figure 5 continued on next page*

*Figure 5 continued*

OTX2<sup>CRISPR</sup> cells showing average reads per cell. (**F**) Pie charts showing the percentages of cells found in each cluster of the CTRL and mutant samples. Bars under the pie charts show distribution of cell cycle markers across the two datasets. (**G**) Heatmap of the differentially expressed genes across the clusters of both CTRL and OTX2<sup>CRISPR</sup> cells. Purple represents low gene expression and yellow represents high expression.

The online version of this article includes the following figure supplement(s) for figure 5:

**Figure supplement 1.** GFP collection gates, expression in the clustered cells and validation of the CRISPR-induced mutations in the analyzed mutant retinas.

**Figure supplement 2.** Representative markers for each cluster in CTRL and OTX2<sup>CRISPR</sup> samples.

**Figure supplement 3.** Heatmap of the differentially expressed genes across the clusters of both CTRL and OTX2<sup>CRISPR</sup> cells.

of the combined control and mutant cell datasets categorized them as a single cluster. However, a clear division was observed on the heat map representation (*Figure 5*). Differential expression of two of the most significant markers that correlated with this demarcation were OTX2 (high in cells on the left half) and PAX6 (high in cells on the right half). It was determined that these sub-groups corresponded to their origin from the mutant or control samples by separating these two groups (*Figure 5—figure supplement 3*). Analysis between the OTX2<sup>CRISPR</sup> and CTRL restricted progenitor populations identified that PAX6 was one of the most differentially expressed gene present (*Supplementary file 6*). In addition, several other genes besides PAX6 that are normally in multipotent RPCs, but are down-regulated in the restricted RPCs, were upregulated in the mutant population. These included SIX3, NOTCH1, ID1, and PROX1 and supports a previously suggested role for OTX2 in mediating a transition from multipotent to restricted RPCs (*Supplementary file 6*), (*Buenaventura et al., 2018*). Moreover, a downregulation of a subset of genes (DIO3, TFAP2D, and OTX2 itself) normally expressed at low levels in multipotent RPCs and upregulated in restricted RPCs was observed, which also supports a positive role for OTX2 in establishment of the restricted RPC state (*Supplementary file 6*).

## One third of the OTX2 mutant cells have a unique molecular signature and RGC-like morphology

The combined analysis of the CTRL and OTX2<sup>CRISPR</sup> mutant cells revealed a new cluster of cells, comprising 34% from the total dataset in the mutant, compared to 0% (17 cells) in the CTRL one.

Initial analysis of the markers suggested that there was a lack of clear-cut classification to a known retinal cell class at any time point, and certainly at the time point of this analysis. These cells express G1/G0 specific markers (*Figure 6—figure supplement 1B,C*) and the most differentially expressed genes compared to other clusters are SPIK2, BRINP2, SULF2, ERBB4 as well as CRABP1. In addition, they express high levels of PAX6, OLIG2, TUBB3, ONECUT3 and low levels of OTX2.

Considering the fact that the OTX2ECR2::GFP reporter showed a marked increase in the cell populations in the inner retina, as well as to exclude the hypothesis that these cells represent cell doublets, since they express biologically incompatible genes like PAX6 and OTX2, the simultaneous expression of several retinal markers, known to be expressed by the cells in the inner retina were analyzed. A qualitative analysis showed that several GFP-positive cells are ISL1/panPOU4F/LHX1-negative in both OTX2<sup>CRISPR</sup> mutants, suggesting that these cells may not be HCs or RGCs (*Figure 7A–F*). Surprisingly, these cells resemble RGCs with a large soma, dendritic-like neurites as well as a long, axonal-like neurite located within the fiber layer and oriented towards the optic nerve head.

To assess the lineage progression of the cells that form this cluster, a recombinase-based approach was used to label cells that show history of OTX2ECR2 activity as described in *Schick et al. (2019)*. Retinas were in ovo electroporated with either empty p18 or OTX2<sup>CRISPR</sup> g2 plasmids, along with a plasmid where the OTX2ECR2 element drives the expression of the PhiC31 recombinase and a responder plasmid, where a*tt* sites flank a Neomycin-STOP cassette, followed by EGFP ORF. Therefore, GFP is expressed in cells that have a history of OTX2ECR2 activation from the time of the electroporation (E3) to the time of analysis (E10), when all cell types are formed and retinal lamination is complete.

OTX2<sup>CRISPR</sup> g2 retinas show an increased number of RGCs confirmed by the morphology of the OTX2ECR2 lineage-traced GFP-positive cells, as well as their panPOU4F immunoreactivity

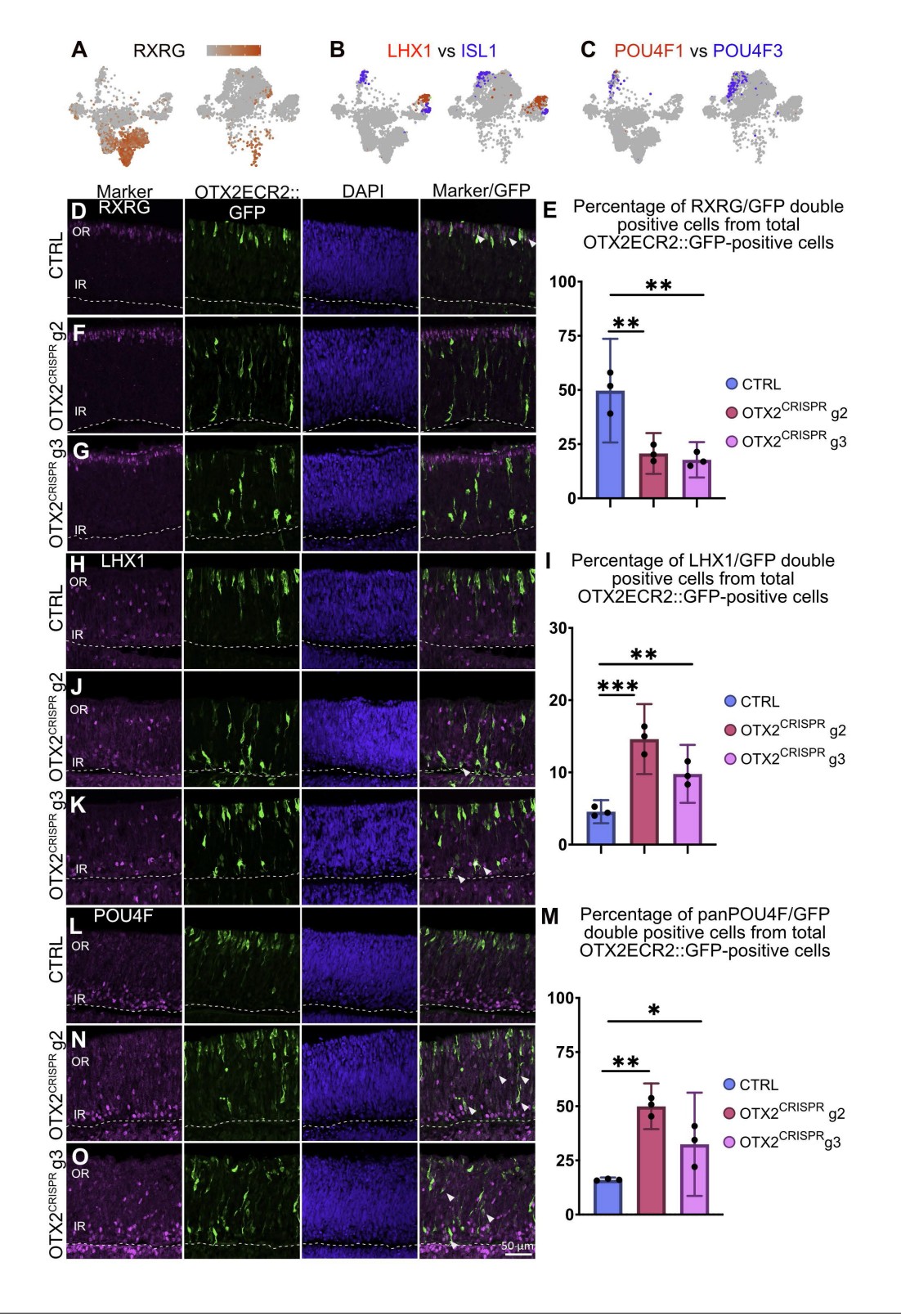

**Figure 6.** Quantitative validation in retinal sections of genes with altered expression in the single cell OTX2 mutation analysis. (**A**) Heatmap in a TSNE view of RXRG in the CTRL (left) and OTX2$^{CRISPR}$ (right). The expression of RXRG is severely reduced in the mutant dataset. (**B**) Comparative analysis of the simultaneous expression of LHX1 (red) and ISL1 (blue) in TSNE plots of CTRL (left) and OTX2$^{CRISPR}$ (right). (**C**) Comparative analysis of POU4F1 and POU4F2 in mutant and control retinas. Similarly, expression of POU4F1 remains restricted to a small number of cells, while POU4F2 (blue) is expressed

*Figure 6 continued on next page*

*Figure 6 continued*

by an increased number of cells in the OTX2$^{CRISPR}$ sample. (D–O) Quantitative analysis of the RXRG, LHX1 and panPOU4F proteins in the CTRL (D, H, L) and both g2 (F, J, M) and g3 (G, K, O) mutant retinas. Bar graphs represent mean percentages of OTX2ECR2::GFP/marker-double positive cells from the total number of OTX2ECR2::GFP cells. Data points represent the biological replicates, n = 3 for each. Error bars represent 95% confidence intervals. OR, outer retina, IR, inner retina.

The online version of this article includes the following figure supplement(s) for figure 6:

**Figure supplement 1.** Mutually exclusive OTX2 and PAX6 expression, cell cycle distribution and POU4F1 colocalization with OTX2ECR2::GFP.

(*Figure 7H–K*). These GFP/POU4F cells appear to have higher levels of POU4F marker, when analyzed qualitatively. Quantification of the OTX2ECR2 lineage-traced cells in the OTX$^{CRISPR}$ mutants showed that 77.6% (st. dev. ±23.34%) express the panPOU4F marker, whereas only 3.48% (st. dev. ±4.9%) express the marker in the CTRL retinas.

Taken together, these data suggest that cells forming cluster 'New' represent cells found in an intermediate state and require additional developing events for their specification into RGCs.

## Discussion

The utility of the CRISPR/Cas9 system in the induction of targeted mutagenesis has been firmly established across multiple paradigms, not only facilitating the study of somatic mutations at any given time during development, but also allowing these studies in less genetically-tractable model organisms, like the chicken (*Gandhi et al., 2017*). Furthermore, the ability to introduce mutations in specific tissues at particular timepoints has proven particularly useful for developmental studies, allowing for a more precise evaluation of genes that have a role in multiple developmental time windows. In addition, the recent technological advances for the generation of single cell gene expression profiles allows for analysis of rapidly dividing and differentiating cells at a single cell resolution (*Carter et al., 2018*; *Zeisel et al., 2018*). The current study illustrates the power of using these methods in combination to evaluate gene function in the context of a developing tissue with notable cellular complexity.

### Timing and CRISPR/Cas9 effectiveness of the OTX2 ablation

Introduction of OTX2$^{CRISPR}$ at multiple timepoints allowed for identification of previously described phenotypes for OTX2, including effects on eye morphology, RPE pigmentation and gene expression, PR and BC formation, and repression of PAX6 expression. Moreover, this current mutation model resembles the clinical manifestations of human mutations of OTX2 that lead to different degrees of microphthalmia and anophthalmia more than any other reported model of OTX2 ablation. Electroporation during eye cup development, even in cases where the electroporation efficiency was relatively low, led to these phenotypes, which suggests that OTX2 expression is highly required during optic vesicle development. In human, similar malformations were reported to occur as an effect of de novo nonsense mutations that lead to early termination of OTX2 transcription and either lead to degradation of the transcript by nonsense-mediated RNA decay or to dysfunctional OTX2 protein due to defects in the homeodomain (*Boyl et al., 2001*; *Gat-Yablonski, 2011*).

The amplicon analysis in this study suggested that there was a high rate of mutagenesis with g2 that either disrupted the sequence of the functionally important homeodomain or created a large truncation. Almost 30% of these reads were wildtype in the targeted region, which given the phenotypic severity of the mutation (for example, only ~5% of photoreceptors remained in the mutant), suggests that either this method underrepresented the number of mutant alleles and/or loss of one copy of OTX2 was phenotypic. The former possibility is unlikely to be caused by nonsense-mediated decay given the proximity of new stop codon usage to the last exon-exon junction in the transcript (*Lindeboom et al., 2016*). However, it could be caused by failure to recover mutations in which one of the primer binding sites was removed or if mutant cells have reduced OTX2 transcription, for example, in response to cell fate changes. In support of this, the percent of OTX2 positive cells in the mutant is lower than in the CTRL samples (46.1% versus 77.9%). It is also possible that cells with only one targeted OTX2 allele may contribute to the observed phenotypes, as haploinsufficiency

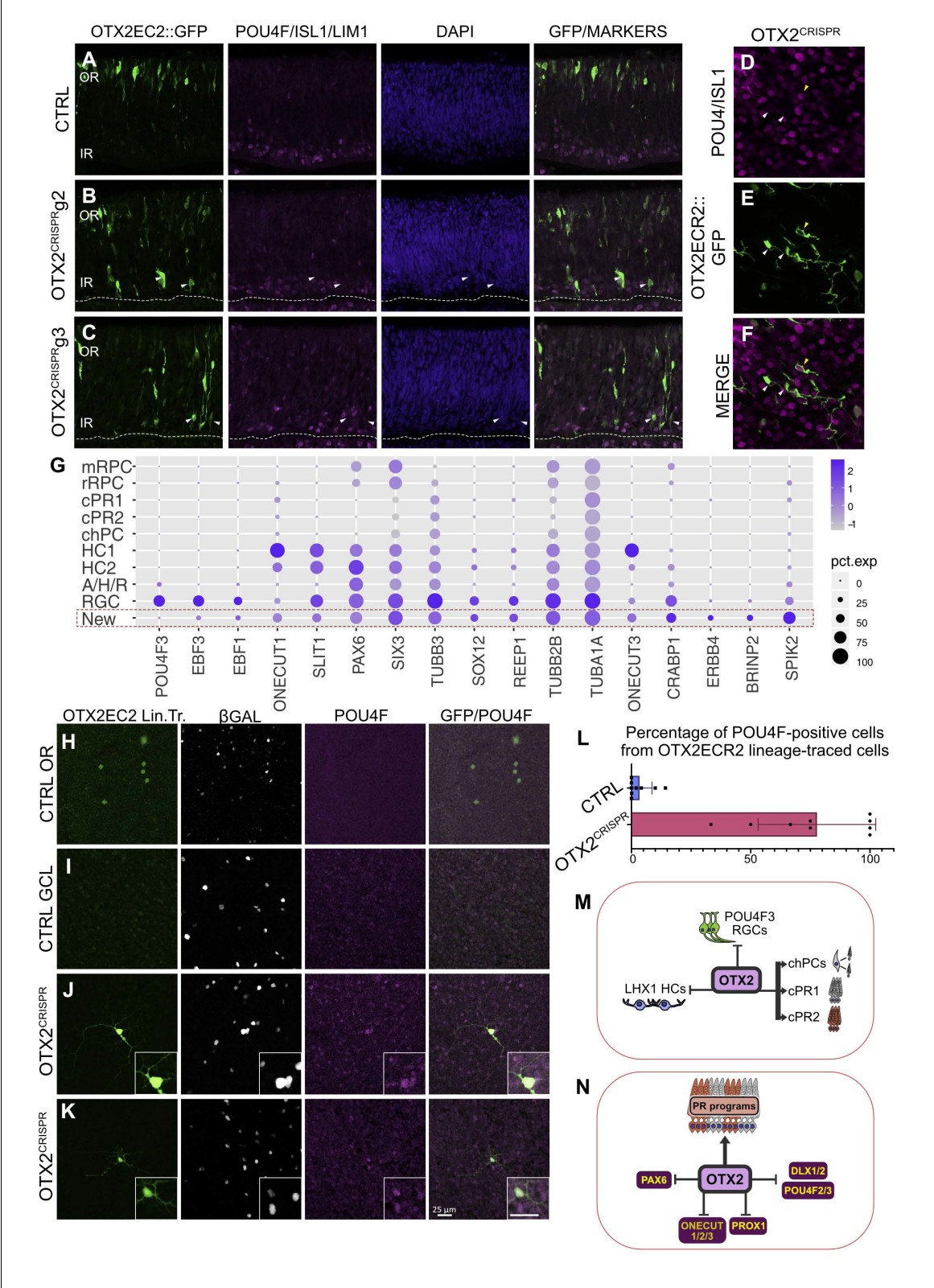

**Figure 7.** OTX2 mutation results in the generation of a cellular population with morphological features of RGCs but that does not express RGC markers at the time of the single cell analysis. (A–C) Confocal micrographs of vertical sections of retinas electroporated with the OTX2ECR2::GFP reporter and respective CRISPR complex (control or with guide) at E5 and cultured for 2 days and immunostained simultaneously with panPOU4F, ISL1 and LHX1 along with GFP. White arrows show GFP-positive cells that are not positive for any of the three markers. (D–F) Confocal micrographs of whole mount

*Figure 7 continued on next page*

*Figure 7 continued*

retinas electroporated with the OTX2ECR2::GFP reporter and respective CRISPR g2 at E5 and cultured for 2 days and immunostained simultaneously with panPOU4F and ISL1, along with GFP. White arrows show cells that are not positive for the markers, while the yellow arrow shows a marker positive RGC. (**G**) Dot plot presenting representative markers for cluster New. The size of the dot corresponds to the percentage of cells expressing the marker (x-axes) in each cluster (y-axes). The blue color intensity represents the average of the expression level. (**H–L**) Lineage tracing of the OTX2ECR2 element shows predominantly RGCs in the OTX2[CRISPR] retina, compared to the PR-rich CTRL. OTX2ECR2 element drives the expression of PhiC31 recombinase, that, upon recombination of the attachment sites, results in GFP expression in all cell with past and present activity of the OTX2ECR2 element. OTX2ECR2 lineage-traced cells in the outermost half of a whole mount CTRL retina (**H**) and in the innermost half of the whole mount OTX2[CRISPR] retinas (**J, K**). (**I**) Micrograph of a CTRL WM retina imaged in the GCL shows no OTX2ECR2 lineage traced cells are amongst the β-Galactosidase (electroporation control marker) positive ones. (**L**) Quantitative analysis of whole retina thickness in both CTRL and mutant samples. Bar graph represents percentages of GFP/POU4F double-positive cells, from total number of GFP-positive cells. Datapoints represent a technical replicate from each retina counted (three technical replicates for each of the three biological replicates). (**M–N**) Contribution of OTX2 during the cell fate specification in the developing chick retina. (**M**) OTX2 is necessary for the generation of two PRs types, as well as one type of cone homotypic progenitor cell (chPCs), repressing the generation of subtypes of RGCs and HCs. (**M**). OTX2 activates genes required for formation of PRs, inhibiting regulators of RGC (POU4F2/3, DLX1/2), HC (ONECUT, PROX1) and PAX6 expression. WM, whole mount, OR, outer retina, GCL, ganglion cell layer.

effects for OTX2 have been observed in both human and mouse studies (*Ragge et al., 2005*; *Wyatt et al., 2008*; *Henderson et al., 2009*; *Kim et al., 2015*).

## Cell type-specific effects in response to loss of OTX2

This study is focused on the OTX2 transcription factor, known to be involved in several aspects of retinogenesis, but for which critical mechanistic details are undefined. The choice of the model organism – chick – not only increases the toolbox of developmental studies, the retina being easily accessible for manipulations during the embryonic time period, but also shows common features with the human retina, in both the physiology of the signal detection and transmission as well as gene networks. The expression of four different opsins in chick PRs that allow for signal detection during photopic conditions, as well as the existence of multiple subtypes of HCs represent features that do not exist in the murine retina. In addition, recent studies identified significant similarities between the developing human and chick retina regarding the expression of particular genes, distinctively expressed in only certain cell types, that are not expressed in the mouse retina (*Lu et al., 2019*).

The use of the OTX2ECR2 reporter allowed the characterization of the OTX2 RPC population and its progeny at high resolution. However, to increase the mutagenesis rate of OTX2 in these cells, the Cas9 protein and guide RNAs were driven by regulatory elements with strong, broad spatial activity (to also target multipotent RPCs that generate OTX2 RPCs). Thus, there could be OTX2-positive cells outside of the OTX2ECR2 population that are affected by the gene editing plasmid, and therefore a formal possibility that these mutated cells have a cell non-autonomous effect on the OTX2ECR2 population, either through changes in secreted factors, or through OTX2 itself, which has been reported to be secreted by cells and taken up by others (*Sugiyama et al., 2008*). However, given the existence of OTX2 mRNA and protein in the majority of the OTX2ECR2 active population and previous transcriptomic analyses showing OTX2 mRNA transcripts largely restricted to the overlapping THRBCRM1 population (see below), the effects observed here are likely due to cell autonomous mechanisms.

Previous analysis in the mouse retina concluded that conditional loss of OTX2 initiated in postmitotic PRs, or throughout the retina using an early neural retina driver, leads to loss of PRs, HCs, and BCs, a concomitant increase in ACs, and no change in RGCs. In contrast, this study identified significant increases in the representation of specific HC and RGC subtypes. While this difference could be due to a mouse/chick distinction, we favor the explanation that three experimental aspects of this study allowed for novel findings: 1) short-term assessment after OTX2 mutation, 2) a targeted cell population that includes OTX2-positive RPCs and 3) single cell analysis to sensitively identify cell types. The three previous developmental studies examined the effects of OTX2 mutation initiated either in post-mitotic PRs, or in the early retina prior to postmitotic cell formation and possibly at a timepoint when OTX2 is still critical for eye specification (*Nishida et al., 2003*; *Sato et al., 2007*; *Omori et al., 2011*). In addition, in two of the previous cases, the evaluation of cell fate largely occurred in the adult, possibly precluding the detection of the cell fate changes observed in this

study. However, several of the gene regulatory changes associated with HC and RGC formation identified in this study were identified in the previous study that examined the OTX2 conditional knockout at postnatal day one (*Omori et al., 2011*). These include upregulation of PROX1, associated with HCs, BCs, AII ACs, and of POU4F2 (BRN3B), expressed in RGCs. In addition, DLX1 and 2 were strongly upregulated. While these gene expression changes were not noted as indicative of cell fate changes, they are in fact consistent with the ones observed here in the chick retina and may indicate an evolutionarily conserved role of OTX2 in repression of gene regulatory networks associated with HCs and RGCs.

To date, it has been suggested that OTX2 acts primarily as a direct activator of transcription in the retina, while a potential repressive function, as well as the targets that mediate repression, are not clear. The above-mentioned studies were conducted mainly in the postnatal retina, missing the early developmental time points where there could be unique gene regulatory networks engaged by OTX2 in restricted RPCs. One of the most striking transcriptional changes to be identified was upregulation of PAX6. While previous studies also noted a robust induction of PAX6 in response to OTX2 loss, this was primarily thought to be linked to the observed increase in amacrine-like cells, which highly express PAX6. The present study shows that cluster AC/HC/RGC, with cells that express PAX6, increases from 3% in the control dataset to 5% in the OTX2$^{CRISPR}$ one. However, the nature of the single cell analysis allows us to observe that in restricted RPCs PAX6 was the gene most upregulated in OTX2$^{CRISPR}$ cells and this is not likely to be a secondary effect from a cell fate change, since the majority of the gene expression signature of these restricted RPCs stays the same. In addition, the same mutually exclusive expression between PAX6 and OTX2 was also noticed in the RPE of the retinas with induced OTX2 mutation at E1.5 (*Figure 1*). This suggests that the causal link between the two is more direct within retinal gene regulatory networks. The mechanism by which OTX2 functions to repress PAX6 is not clear as of yet. Previous studies have identified a role for a physical interaction of OTX2 with TLE4 that allows for direct transcriptional repression by OTX2 occupancy (*Agoston and Schulte, 2009*). Alternatively, repression could be mediated by a transcriptional target of OTX2 such as the transcriptional repressor PRDM1 (*Wang et al., 2014*, *Mills et al., 2017*). Previous observations from our group showed that OTX2, when misexpressed with ONECUT1, was able to drive formation of a restricted RPC population from a multipotent population (*Buenaventura et al., 2018*). Whether the effect observed here on PAX6, a functionally important transcription factor in multipotent RPCs, could contribute to this effect, is currently unknown.

In addition to the increase in the percentages of cells that form clusters RGC and HC1 in the OTX2$^{CRISPR}$ dataset, the generation of a new cluster, with cells that do not appear to have the clear transcriptional signature of any single final cell type in the retina, represented one of the novel findings of this study. These cells also express PAX6, are located in the inner retina and show predominantly RGC-like morphologies. However, they do not express high levels of any markers used for RGC identification. Considering their morphology, as well as the fact that the OTX2ECR2 lineage tracing shows that predominantly all GFP positive cells appear to be panPOU4F RGCs, it is possible that, at the time of the single cell analysis, these cells were in a transitional state, prior to activation of RGC transcriptional programs in the following stages. Alternatively, some of these cells could undergo apoptosis, as the number of the OTX2ECR2 lineage traced cells was reduced in comparison with the CTRL retinas analyzed.

While cell fate choice in the vertebrate retina has been an area of intense study, a coherent model of the fundamental molecular and cellular mechanisms underlying this process is still lacking. One question is whether fates are decided according to stochastic or deterministic mechanisms. Several studies that examined zebrafish random progenitor cell clones have suggested that stochastic mechanisms are the primary drivers of postmitotic cell fate choice (*Boije et al., 2015*; *He et al., 2012*). However, studies in zebrafish, chicken, and mice that have used cis-regulatory elements to genetically target specific RPC populations have identified what appear to be non-random preferences for postmitotic fate choices. This suggests that there are some deterministic factors that inform these decisions. In this study, a subpopulation of the restricted RPC types labeled by the OTX2ECR2 element is the RPC population defined by the THRBCRM1 regulatory element. These progenitor cells preferentially form cones, type1 horizontal cells, and a small population of RGCs over other cell types such as rods and type 2–4 horizontal cells (*Schick et al., 2019*). Experiments comparing the overlap between the two lineages showed an overlap of 32.9% c.i. ±9.5 of OTX2ECR2 cells that are THRBCRM1 (MGT). Interestingly, we observed that loss of OTX2 led to a loss of photoreceptors and

a concomitant increase in type1 HCs and non-POU4F1 positive RGCs. These specific choices suggest that deterministic factors could influence the choices of mutant OTX2 RPCs as well. We speculate that this is related to the fact that type 1 HCs and possibly non-POU4F1 RGCs are already preferred fate choices of THRBCRM1 RPCs. We have noted previously that these RPCs are defined by OTX2 and ONECUT1 co-expression and that these two transcription factors become individually expressed in the postmitotic daughter cells that become cones and HCs, respectively. All members of the ONECUT family are upregulated in OTX2$^{CRISPR}$ retinas, suggesting that these factors appear to be derepressed in the cells that would normally be photoreceptors and, therefore, drive HC genesis.

Taken together, the current study proposes a model in which OTX2 serves as a key positive regulator of photoreceptor genesis from restricted RPCs, while repressing specific subtypes of other retinal fates (*Figure 7M*). At the gene regulatory network level, OTX2 represses key transcription factors involved in non-photoreceptor cell types (*Figure 7N*). Further combined use of the single cell sequencing/CRISPR gene editing approach with variation of time, targeted genes, and labeled cell populations will provide a powerful genetic strategy to examine developmental gene regulatory networks.

## Materials and methods

**Key resources table**

| Reagent type (species) or resource | Designation | Source or reference | Identifiers | Additional information |
|---|---|---|---|---|
| Genetic reagent (*Gallus gallus*) | GRCg6a | International Chicken Genome Consortium | GCF_000002315.5 | |
| Transfected construct | Modified px330: P18/MG18 | Modified from the *Cong et al., 2013* version | | The original promoter of the px330 upstream of the Cas9 ORF was replaced with the CAG promoter |
| Transfected construct | Modified px330 with OTX2 g2: MG10 | Lab made | | |
| Transfected construct | Modified px330 with OTX2 g3: MG233 | Lab made | | |
| Transfected construct | Modified Stagia3: ME1860 | *Emerson and Cepko, 2011* | | Mouse OTX2 ECR2 EGFP reporter as described in *Emerson and Cepko (2011)* |
| Transfected construct | CAG::βGal | Cepko lab | | Expression vector, nuclear βGalactosidase reporter driven by the CAG promoter |
| Transfected construct | CAG::EGFP | *Matsuda and Cepko, 2004* | RRID:Addgene_11150 | Expression vector, EGFP reporter driven by the CAG promoter |
| Transfected construct | CAG::iRFP | *Buenaventura et al., 2018* | | Expression vector, iRFP reporter driven by the CAG promoter |
| Antibody | Anti-OTX2 Goat polyclonal | AF1979 | RRID:AB_1617988 | |
| Antibody | Anti- OTX1+OTX2 Rabbit polyclonal | ab21990 | RRID:AB_776930 | |
| Antibody | Anti-PAX6 Mouse IgG1 monoclonal | PAX6 | RRID:AB_528427 | |
| Antibody | Anti-BRN3A Mouse IgG1 monoclonal | Mab1585 | RRID:AB_94166 | |
| Antibody | Anti-panBRN3 Mouse IgG1 monoclonal | sc-390780 | | |

*Continued on next page*

*Continued*

| Reagent type (species) or resource | Designation | Source or reference | Identifiers | Additional information |
|---|---|---|---|---|
| Antibody | Anti-LIM1 Mouse IgG1 monoclonal | 4F2-C | RRID:AB_531784 | |
| Antibody | Anti-ISL1 Mouse monoclonal | 39.3F7 | RRID:AB_1157901 | |
| Antibody | Anti-PH3 Rabbit polyclonal | 06–570 | RRID:AB_10582726 | |
| Antibody | Anti-VSX2 Sheep polyclonal | x1180p | RRID:AB_2314191 | |
| Antibody | Anti-βGAL Chick polyclonal | ab37382 | RRID:AB_307210 | |
| Commercial assay | Click-iT EdU Alexa Fluor 647 imaging kit | Invitrogen | Cat#C10340 | |
| Reagent | O. C. T. Compound | Sakura Tissue-Tek | Cat#4583 | |
| Reagent | Fluoromount-G | Southern Biotech | Cat#0100–01 | |
| Reagent | Papain | Worthington | Cat#L5003126 | |
| Other | Raw matrix files | NCBI Gene Expression Omnibus | GEO: GSE142244 | |
| Software, algorithm | SnapGene | | GSL Biotech; snapgene.com | |
| Software, algorithm | Fiji | *Schneider et al. (2012)* | https://fiji.sc/ | |
| Software, algorithm | Affinity Designer | Affinity software | | |
| Software, algorithm | JASP 0.9.0.1 | | | |
| Software, algorithm | GraphPad Prism | GraphPad software | https://www.graphpad.com/scientific-software/prism/ | |
| Software, algorithm | R | R Core Team | https://www.r-project.org/ | |
| Software, algorithm | Seurat | *Macosko et al. (2015)* | http://satijalab.org/seurat/ | |
| Software, algorithm | ChopChop | *Montague et al. (2014)*; *Labun et al. (2016)* | https://www.chopchop.cbu.uib.no | |
| Software, algorithm | FlowJo 10.2 | | https://www.flowjo.com | |
| Sequence-based reagent | Sex det. forward | This paper | PCR primers | CCCAAATATAACACGCTTCACT |
| Sequence-based reagent | Sex det. reverse | This paper | PCR primers | GAAATGAATTATTTTCTGGCGAC |
| Sequence-based reagent | Sex det. Control region forward | This paper | PCR primers | AGCTCTTTCTCGATTCCGTG |
| Sequence-based reagent | Sex det. Control region reverse | This paper | PCR primers | GGGTAGACACAAGCTGAGCC |
| Sequence-based reagent | Amplicon sequencing forward | This paper | PCR primers | GGAGCGCACCACCTTCAC |
| Sequence-based reagent | Amplicon sequencing reverse | This paper | PCR primers | CTGCACTCTGGACTCGGG CTGCACTCTGGACTCGGG |

## Experimental model and subject details

All experimental procedures were carried out in accordance with the City College of New York, CUNY animal care protocols. Fertilized chick eggs were obtained from Charles River, stored in a 16° C room for 0–10 days and incubated in a 38°C humidified incubator. Experiments were performed

on unbiased number of female and male animals. For the scRNA sequencing experiment only, retinas from embryos PCR-identified as female were processed, to avoid variability from the reference genome.

For sex determination, genotyping PCRs were completed as explained in *Clinton et al. (2001)*. Briefly, the technique is based on the presence of *Xho* I repeats, a class present on the W chromosome. A PCR reaction using primers flanking that region (forward 5' CCCAAATATAACACGCTTCAC T 3'; reverse 5' GAAATGAATTATTTTCTGGCGAC 3') as well as primers flanking a control region (forward 5' AGCTCTTTCTCGATTCCGTG 3', reverse 5' GGGTAGACACAAGCTGAGCC 3') were used. The presence of both bands identifies DNA of female origin, while presence of only the control gene band marks DNA of male origin.

Experiments were done in non-randomized, non-blinded conditions with the exception of quantitative data in *Figure 6* that involved manual counting of images by an observer blinded with respect to a sample's origin as experimental or control.

## Method details

### CRISPR/Cas9-induced mutation design

To determine guide RNA sequences optimal for the ablation of the OTX2 gene www.chopchop.cbu. uib.no online tool was used, and the sequence of the first two coding exons of the gene were analyzed (*Montague et al., 2014*; *Labun et al., 2016*). The target DNA sequence with the best score calculated according to the online algorithm were selected and processed further.

The sequence of each guide RNA, excluding the PAM motif, was cloned into a modified px330 vector (Addgene number #42230) between the two *Bbs*I sites downstream the U6 promoter, as described in *Cong et al. (2013)*. In the px330 vector, the CBh promoter was replaced with the CAG synthetic promoter (*Okabe et al., 1997*), using *Xba*I *Age*I restriction sites upstream of the humanized *S. pyogenes* Cas9, herein called p18. Following plasmid delivery into target cells, Cas9 protein induces a double-stranded break which is resolved through the error-prone DNA repair mechanism, therefore genomic modifications were obtained through the non-homologous end joining pathway (NHEJ). The sequences of the two guides used are – g2: GGCGCAGCTGGACGTGCTGG<u>AGG</u>, g3: GATTTGTTGCATCCGTCCGT<u>CGG</u>, the underlined letters represent the PAM motif.

Expression vectors containing either fluorescent reporters or nuclear LacZ driven by the CAG promoter were used as electroporation controls (*Matsuda and Cepko, 2004*; *Buenaventura et al., 2018*; *Emerson and Cepko, 2011*). For the CAG::tdT reporter, the CAG promoter was cloned with *Sal*I *EcoR*I in the STATIA vector. This represents a modified version of the STAGIA3 plasmid, where EGFP was replaced with TdTomato using *Age*I *Bsr*GI restriction sites.

The lineage tracing was done as described in *Schick et al. (2019)*. Briefly, the OTX2ECR2 sequence was cloned upstream the open reading frame of the PhiC31 recombinase. This plasmid was electroporated simultaneously with a different one where *att* recombination sites that flank a Neomycin-STOP cassette, followed by the GFP sequence. Upon recombination of the *att* sites by the PhiC31 recombinase, GFP is expressed in cells with a present or past activity of the OTX2ECR2 enhancer, therefore allowing the analysis of these cells further in development.

## Electroporation and explant culture

In vivo electroporation was performed at two developmental time points E1.5 (HH stage 9–11) and E3 (HH 18) on healthy-developed embryos. At E1.5, plasmids and Fast Green tracer (0.1% final concentration) in TE buffer was injected into the optic vesicle using a pulled glass needle. A sharp negative electrode was inserted into the forebrain at equal distance between the optic vesicles, while a mobile, positive electrode, was placed near the exterior side of the optic vesicle.

For E3 electroporations, DNA-dye mixes were injected into the subretinal space. The negative electrode penetrated the head in proximity to the eye's dorsal side and the positive electrode was placed frontal to the eye. For both types of in vivo electroporations DNA cocktails contained the reporter plasmid used as electroporation control along with the p18 vector, with or without guide RNA sequence, all at a concentration of 2 µg/µl. Three pulses of 10V were applied using a Nepagene Super Electroporator NEPA21 Type II electroporator. Following the procedure, the windowed eggs were covered with transparent tape and returned to the 38°C incubator.

Ex vivo electroporations were done as previously reported in *Buenaventura et al. (2018)*. Briefly, prior to electroporation the retinas were dissected into warm DMEM/F12 media (Life Technologies, 11320082). Plasmids of interest diluted in 1X PBS for a total volume of 50 µl filled the electroporation cuvette where the retina was placed, with the lens surface attached to the negative electrode. Five 50 ms pulses of 25V with a 950 ms interpulse interval were applied using the same electroporator. In general, plasmids were used at following concentrations - 100 ng/µl for the reporter plasmids driven by the CAG promoter, 160 ng/µl for enhancer-driven reporter plasmids and 200 ng/µl for Cas9 based plasmids. After electroporation, lenses were dissected out and retinas were placed on porous filters of 0.2 µm (13 mm Nuclepore Track-Etch Membrane Whatman filters) floating on 1 ml basic culture media, 10% FCS, 1X Pen/Strep, 1X L-glutamine in DMEM/F12 (Life Technologies, 10378016) and incubated at 37°C with 5% $CO_2$.

## Immunohistochemistry

Retinas processed for immunohistochemistry were fixed in 4% paraformaldehyde for 30 min at room temperature, sunk in 30% sucrose and snap-frozen in OCT (Sakura Tissue-Tek, 4583). Vertical sections of 20 µm were obtained using a Leica Cryostat and collected on microscopy slides (Fisher-Brand, 12-550-15).

Immunofluorescence protocol was as described in *Buenaventura et al. (2018)*. A list of all primary antibodies used, along with their vendor information and concentration used is shown in *Supplementary file 1*. Following incubation with the primary antibody, retinal sections were washed in 1X PBS, 0.1% Tween for a total of 30 min at room temperature, then blocked for additional 30 min in the blocking solution described above. Secondary antibodies used were: Alexa 488 and Alexa 647 to a 1:400 dilution (Jackson Immunoresearch, 115-605-207), and Cy3 to a 1:250 dilution (Jackson Immunoresearch 115-165-205). All secondary antibodies were used in different combinations according to the host of the primaries (anti-rabbit, anti-mouse, anti-sheep and anti-goat) and were appropriate for multi-labeling. For nuclear counterstaining a solution of 4′,6-diamidino-2-phenylindole (DAPI) in 1X PBS was applied on sections prior to three final washes of 15 min at room temperature in 1X PBS. Slides were mounted in Fluoromount (Southern Biotech, 0100–01) with 34 × 60 mm cover slips (VWR, 48393 106).

To immunofluorescently label dissociated cells in suspension, the following procedure was applied – after the 15 min fixation retinas were washed with 1X PBS, followed by washes in 1X PBS, 0.1% Tween. Cells were blocked for 1 hr in 5% normal serum of the species the secondary antibodies were raised in, then incubated overnight at 4°C in a solution containing the primary antibodies and 5% serum. The next day, cells were washed with 3 ml 1X PBS, 0.1% Tween and blocked in 5% serum for 30 min. Secondary antibodies were added to the blocking solution to the concentrations noted above and incubated for 1 hr at room temperature protected by light.

For EdU incorporation in vivo, a solution of 10 µM in 1X PBS was added on top of the embryo 5 hr prior to sacrifice (*Warren et al., 2009*). To develop, a Click-iT EdU Alexa Fluor 647 imaging kit was used in addition to the regular immunostaining protocol (Invitrogen, C10340).

## Microscopy and figure design

High magnification images were acquired with an Axiozoom V16 microscope using a PlanNeoFluor Z 1x objective at a digital zoom of 37.5.

Confocal micrographs were acquired at a 1024 × 1024 resolution using an inverted Zeiss LSM710 confocal microscope with an EC Plan-Neofluar 40x/1.30 Oil DIC M27 objective. During acquisition a ZEN software was used. All images were processed and converted into .tiff format using the FIJI version of ImageJ (*Schneider et al., 2012*). Figures were assembled using Affinity Designer vector editor; general brightness and contrast adjustments were done when necessary on the entire field imaged. The SnapGene software was used for generation of the schematics in *Figure 1A, B Figure 3A* and *Figure 5—figure supplement 1F and H*.

## Dissociation and cell sorting

Retinal explants were dissociated after being cultured for 48 hr with a papain-based protocol (Worthington, L5003126) as described in *Jean-Charles et al. (2018)*. After dissociation, cells were fixed in

4% PFA at room temperature for 15 min, washed and analyzed, or immunostained as described above, prior to analysis.

The fluorescent activated cell sorting (FACS) of the GFP reporter was done using a BD FACS Aria machine with a 488 nm laser. For sorting experiments requiring high cell viability post sorting and nucleic acid extractions, cells were resuspended and sorted in cold DMEM 10% FBS media (Gibco 10437–010).

## Library preparation, sequencing and data analysis

Single cell RNA sequencing of OTX2ECR2::GFP positive cells in the OTX2CRISPR mutants and CTRL conditions, 40,000 GFP-positive cells from four retinas per sample were collected into DMEM 10% FBS media, pelleted at a speed of 500 *xg*, then resuspended in the same media. A trypan blue exclusion viability test confirmed that the percentage of viable cells was greater than 70%. Approximatively 12,000 cells were input into the 10X Genomics Chromium machine and individual droplets were formed containing cell and molecule specific barcodes. Libraries were sequenced on an Illumina 4000 sequencer. The data alignment on the GRCg6a reference genome, barcode processing and generation of cell-gene matrices was done at the Single Cell Analysis core facility at Columbia Genome Center using the 10X Genomics pipeline. Matrices were further analyzed with R using the Seurat package (*Satija et al., 2015*). In total, 5,122 cells were recovered from the CTRL sample and sequencing yielded an average of 65,555 reads per cell, with a total number of 17,485 genes detected and a median of 1293 genes per cell. From the OTX2CRISPR sample, 4,940 cells were recovered, sequencing yielded an average of 66,303 reads per cells with a total of 17,351 genes detected and a median of 1254 genes per cell.

For the analysis, the dataset obtained from the OTX2 CTRL retinas was processed individually, as well as in combination with the OTX2CRISPR dataset. Deatiled R scripts for the analysis of both datasets are found in *Source code 1* and *Source code 2*, respectively. Briefly, for both CTRL and combined projects, from the mitochondrial and ribosomal genes, genes detected and unique molecular identifiers (UMIs) outliers were filtered out from the downstream analysis. Cells were assigned cell cycle phases based on their gene expression, and the cell cycle genes were also regressed out. Genes used for the mitochondrial, ribosomal and cell cycle regressions are shown in *Supplementary file 3*.

Data was normalized and scaled, and principal component analysis was performed based on a list of variable genes previously computed.

To classify the cells in the OTX2ECR2 lineage in the CTRL scenario, filtered cells were run through a linear dimensional reduction algorithm using highly variable genes in Seurat (Methods), computing 33 principal components (PCs). Mitochondrial, ribosomal and cell cycle genes were regressed from the analysis as discussed in the methods section (*Supplementary file 3*). All computed PCs were analyzed with the JackStraw procedure (*Macosko et al., 2015*) and 22 were processed further for cluster analysis. The FindClusters function was used in Seurat to compute the clusters, which were then visualized with the t-distributed stochastic neighbor embedding (tSNE) algorithm.

For the 'combined' analysis 40 total PCs were computed initially and 33 were processed further. All 10 clusters assigned in the analysis of the CTRL cells were detected when both datasets were analyzed simultaneously.

For the amplicon sequencing, total mRNA was extracted using a Qiagen RNeasy Mini kit (Cat. No. 74104) from cells from the sorting experiment used to generate the single cell analysis. The reverse transcription to obtain complementary DNA was done using the qScript cDNA Super Mix, Quanta bio kit (Cat. No. 95048–025) and the cDNA was used as template for the PCR amplification of the OTX2 region, flanked by the forward – 5' GGAGCGCACCACCTTCAC 3' and reverse – 5' C TGCACTCTGGACTCGGG 3' primers, both containing the Illumina adapters: 5' ACACTCTTTCCC TACACGACGCTCTTCCGATCT 3', for forward sequencing read and 5' GACTGGAGTTCAGACGTG TGCTCTTCCGATCT 3', for the reverse sequencing read. Amplicons with attached adaptors were submitted to Genewiz for Amplicon EZ paired-end sequencing on an Illumina platform. Data analysis was done by the provider using the NGS Genotyper v1.4.0 software.

## Quantification and statistical analysis

Quantitative analysis of the fluorescent reporters and of the antibody-detected markers was done with a BD LSRII machine, using the 488 nm, 561 nm and 633 nm lasers. The analysis was carried out with the FlowJo Version 10.2 software. Quantifications done to validate the results of the transcriptome analysis were done manually, by counting a minimum of three technical replicates for each of the three biological replicates.

In each experiment for quantitative analysis, a minimum of four biological replicates were used in at least three technical replicates. Comparisons between OTX2$^{CRISPR}$ mutants and control groups were done using student's *t*-test with independent samples when two groups were compared, and one-way ANOVA when both OTX2$^{CRISPR}$ g2 and g3 were analyzed in the same comparison with the CTRL. Shapiro-Wilk normality test and Levene's equality of variances assumptions confirmed the normal distribution of the data. Dunnett Post Hoc comparison was done for the analysis of the CTRL, g2 and g3 quantifications. Dunn's nonparametric test was used for the group where data didn't fit in a gaussian distribution curve. For each quantification of CTRL vs one of the two OTX2$^{CRISPR}$ mutants, one eye was used as experimental (OTX2$^{CRISPR}$), while the contralateral represented the control. Error bars represent 95% confidence intervals (c.i.), with the exception of the lineage tracing quantification, for which standard deviation was used. For all experiments statistical analysis was done using JASP 0.9.0.1, GraphPad - PRISM and Microsoft Office Excel 16 software.

For the quantification of the gross anatomical defects induced after electroporation of the CRISPR elements at E1.5 the analysis was done at E5. Both the electroporated eye (right) and the un-electroporated one (left) were dissected, removing the additional tissue surrounding the RPE. The eyes were imaged from both frontal (lens side) as well as dorsal (optic nerve side), area of the eye was measured, then averaged between the two sides. The measurements were reported as mean ratios between the average of the right eye area and the left one (numbers shown in *Supplementary file 2*).

For quantifications done to validate the results showed by the transcriptome analysis using the RXRG, LHX1, panPOU4F (*Figure 6*) and POU4F1 antibodies (*Figure 6—figure supplement 1*), as well as for the lineage tracing quantification (*Figure 7*), percentages represent number of OTX2ECR2::GFP/marker-double positive cells from the total number of OTX2ECR2::GFP-positive cells.

For the transcriptome analysis, three retinas were pooled for each of the CTRL and OTX2$^{CRISPR}$ samples; for each of the mutant retinas included in the sample, the contralateral eye was used as a CTRL to limit the variability across individual animals.

## Acknowledgements

Jeffrey Walker and Jorge Morales provided excellent technical support with flow cytometry and confocal microscopy experiments. We thank the members of the Emerson lab for support throughout the project, Sruti Patoori for the suggestions on statistical analysis and Cosmin Tegla, Revathi Balasubramanian and Justin Brodie-Kommit for critically reading the manuscript.

## Additional information

### Funding

| Funder | Grant reference number | Author |
|---|---|---|
| National Eye Institute | R01EY024982 | Mark M Emerson |
| National Institute of General Medical Sciences | T34GM007639 | Kevin C Gonzalez |
| National Institute on Minority Health and Health Disparities | 3G12MD007603 | Miruna Georgina Ghinia Tegla<br>Diego F Buenaventura<br>Diana Y Kim<br>Cassandra Thakurdin<br>Kevin C Gonzalez<br>Mark M Emerson |

The funders had no role in study design, data collection and interpretation, or the decision to submit the work for publication.

## Author contributions
Miruna Georgiana Ghinia Tegla, Conceptualization, Data curation, Formal analysis, Supervision, Investigation, Visualization, Methodology, Writing - original draft, Writing - review and editing; Diego F Buenaventura, Formal analysis, Visualization, Methodology, Writing - review and editing; Diana Y Kim, Formal analysis, Investigation, Writing - original draft; Cassandra Thakurdin, Validation, Investigation, Validated the OTX2 antibody and performed dissociation/staining for flow cytometry as well as confocal imaging; Kevin C Gonzalez, Investigation, Constructed CRISPR guide plasmids and carried out electroporation experiments; Mark M Emerson, Conceptualization, Supervision, Funding acquisition, Investigation, Methodology, Writing - original draft, Project administration, Writing - review and editing

## Author ORCIDs
Miruna Georgiana Ghinia Tegla (iD) https://orcid.org/0000-0001-6898-1886
Mark M Emerson (iD) https://orcid.org/0000-0002-1914-5782

## Decision letter and Author response
Decision letter https://doi.org/10.7554/eLife.54279.sa1
Author response https://doi.org/10.7554/eLife.54279.sa2

# Additional files

## Supplementary files
• Source code 1. R script for the analysis of the single cell RNA-sequencing on the CTRL sample.

• Source code 2. R script for the analysis of the single cell. RNA-sequencing on the combined analysis of the CTRL and OTX2$^{CRISPR}$ datasets.

• Supplementary file 1. List of antibodies used in the study.

• Supplementary file 2. Detailed values used in the quantifications, including statistical analysis.

• Supplementary file 3. List of genes regressed in the single cell analysis.

• Supplementary file 4. Markers used for the assignment of the clusters in the CTRL dataset.

• Supplementary file 5. Markers used for the assignment of the clusters in the combined analysis of the CTRL and OTX2$^{CRISPR}$ datasets.

• Supplementary file 6. Markers expressed in the restricted RPC cluster.

• Transparent reporting form

## Data availability
Transcriptome data obtained during the current study in matrices format for both CTRL and OTX2-CRISPR is available in the GEO database under GSE142244. Scripts used for data analysis can be found in Source code 1 and 2.

The following dataset was generated:

| Author(s) | Year | Dataset title | Dataset URL | Database and Identifier |
|---|---|---|---|---|
| Ghinia TMG, Emerson MM | 2020 | OTX2 represses sister cell fate choices in the developing retina to promote photoreceptor specification | https://www.ncbi.nlm.nih.gov/geo/query/acc.cgi?acc=GSE142244 | NCBI Gene Expression Omnibus, GSE142244 |

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
