## [Decision Letter]

**Acceptance summary:**

Otx-2 is an important gene that has been implicated in many developmental processes in the brain and in particular in the retina. Its use through multiple developmental times made it difficult to investigate its late role in the retina. Your investigations of these functions using precise CRISPR-based manipulations provide an important contribution to understanding the role of Otx-2 in the visual system and in particular its conserved role in photoreceptor specification,

**Decision letter after peer review:**

Thank you for submitting your article "OTX2 represses sister cell fate choices in the developing retina to promote photoreceptor specification" for consideration by *eLife*. Your article has been reviewed by three peer reviewers, and the evaluation has been overseen by a Reviewing Editor and Marianne Bronner as the Senior Editor. The reviewers have opted to remain anonymous.

The reviewers have discussed their various requests and came out with the list below of experiments/changes.

Major comments:

1) The mutation induced by Cas9 over time needs to be better characterized as providing information on the outcome of the Cas9 editing over time would be valuable data. You should evaluate the percentage of homozygous and heterozygous cells, preferably with respect to the unique indel types. Analyzing RNAseq data from populations of cells as shown in Figure 5—figure supplement 1 might be sufficient: The data show that most reads have indels, which could be useful for determining the percentage of homozygous and heterozygous cells and the indel types). This is important as the study is based on analyses of the outcome of a heterogeneous population of mutant cells that differ in the size and time of deletion.

2) You should also complete the analysis of loss of Otx2 at earlier stages.

3) The use of reporter activity to monitor cell lineage could be misleading as the mutation was induced in all of the transfected retinal progenitor cells, and not only those in which this enhancer is active. The impact of the mutation (autonomous or cell non-autonomous) on the enhancer can distort the conclusion about the fate of the mutant cells, thereby complicating the interpretation of the relevance to Otx2 activity. You should discuss this limitation in the Discussion.

Other comments:

– Referring to the control as "WT" is confusing. Note that you write that the "contralateral non-electroporated eye developed normally". This is not the control mentioned earlier. An accurate term should be provided for the relevant controls – those that were electroporated with Cas9 and guide scaffold. Were the control eyes for the single-cell sequencing electroporated with Cas9 or not? This needs to be clarified.

– Figure 1F: show a higher magnification of the mutant retina and RPE. Not clear if Otx2 is indeed reduced in the RPE.

– Figure 1: also label the region encoding the homeodomain in Figure 1A. In the top panel (F-H), indicate the stages on the right (E1.5-E5), as was done for the lower panel.

– Subsection “CRISPR Cas9-induced ablation of the OTX2 gene at the optic vesicle stage results in anatomical defects of the overall eye structure, pigment epithelium and neuronal retina”: revise the text to eliminate redundancy.

Reviewer #1:

In the present manuscript, Ghinia Tegla and colleagues have investigated the role of OTX2 during retinal cell fate specification. It is well known that OTX2 plays key roles in different processes during CNS patterning, eye morphogenesis, as well the in the terminal differentiation of photoreceptors and bipolar cells. Because of the important early functions of OTX2, null mutants have severe defects and die embryonically. Several studies on the role of OTX2 in postmitotic cells have been published using specific CREs but the specific role of OTX2 in retinal progenitors has remained unresolved.

Here, by means of a systematic OTX2 ablation using CRISPR/Cas9 electroporation in chicken embryos at different stages of development, the authors elegantly address the functions of OTX2 in progenitor cells as well as its genetic networks using single cell RNA-seq. Surprisingly, the authors observe a specific increase in subpopulations of RGCs and Horizontal cells in the absence of OTX2. The work constitutes a meticulous survey of roles of OTX2; it is very well designed, and includes meaningful datasets that shed light onto some unresolved questions. I anticipate that this paper will be a useful resource in the field.

Reviewer #2:

The manuscript details a methodical approach using chick to ascertain the function of OTX2 in retinal development. The authors use CRISPR/Cas9 to target OTX2 in vivo, followed by FACS and SC-Seq to determine gene expression changes. The methods are rigorous and the sometimes complex data are presented in a clear and systematic fashion. The results support the major conclusions of the paper, namely that OTX2 not only is essential to activate the photoreceptor cell fate but also to repress alternative fates. The statistical analyses are well described and appropriate.

Reviewer #3:

Otx2 is one of the most extensively studied transcription factors in forebrain and eye development. Genetic dissection using Cre/loxP in mice has been performed by several groups, leading to the overall notion that this gene is required for photoreceptor, horizontal and bipolar cell fates while inhibiting amacrine cell fate. However, because of its complex expression pattern and the hallmark of the retina where different cells are generated at the same time, there is eventually limited insight on roles of the gene in subsets of progenitors.

The aim of the study by Ghinia Tegla and colleagues is to further explore and decipher the role of Otx2 at different stages and in subsets of retinal progenitors during chick retinal development. To this end, the researchers combined the cutting-edge technologies of in-vivo somatic mutation using the CRISPR/Cas9 gene-editing approach and single-cell RNA-Seq analysis to characterize, at single-cell resolution, the outcome of Otx2 mutation at three stages of development: optic vesicle (during specification of the retinal pigmented epithelium (RPE) and retina, E1.5), optic cup (electroporation of the retina only, E3) and during later stages of retinogenesis (E5).

The first two stages were analyzed qualitatively. The analyses supported reduction of Otx2 in the electroporated cells. The reduction was more extensive at E10, implying that it takes a few days to achieve efficient mutation in many cells. The observed phenotypes served as proof of concept for efficient mutagenesis, and showed that the mutations in chicks resemble the outcome of loss of Otx2 in mammals that suffer from a range of ocular abnormalities, including anophthalmia and coloboma.

Detailed cellular analysis of the phenotype was performed following mutation of Otx2 at E5, along with analysis of the explants 2 days later. In this setting, the authors reported a reduction in Otx2 and in photoreceptor gene expression based on the activity of a reporter that is regulated by Otx2 and is active in cone photoreceptors in the chick retina (THRBCRM2 cis-regulatory element). Next they focused on the impact of the mutation on a subpopulation of retinal progenitors in which one of the Otx2 enhancers has been found by this group to be active (ECR2) in a subset of retinal progenitor cells destined to horizontal cell and cone photoreceptor cell fates. The cells in which ECR2 was expressed were isolated from control and mutant retinas and examined for gene expression using single-cell-seq analyses. The single-cell analysis of transcriptomic changes following the mutation revealed that Otx2 functions to inhibit alternative cell fates, as increases in HZC and amacrine gene expression were detected in the mutant cells. Another finding was the detection of an aberrant type of ganglion-like cell following the mutation. The functional significance of the latter observation is currently not clear.

Overall, this study makes important technical advances as it implements new technologies to functionally address the role of an important developmental regulator in retinogenesis, providing cellular resolution for the functions of this complex gene. However, despite the important technological advances, the overall insights mostly correspond with the known roles of Otx2 in the retina. The following comments need to be addressed to clarify the technological aspects and the interpretation of the findings.

Comments:

1) The mutation induced by Cas9 over time needs to be characterized with respect to the indel fraction, unique indel types, and percentage of homozygous and heterozygous cells. This is important as the study is based on analyses of the outcome of a heterogeneous population of mutant cells that differ in the size and time of deletion (see, for example, analyses presented in Figure 1 of the paper https://www.ncbi.nlm.nih.gov/pubmed/28985529).

2) The use of reporter activity to monitor cell lineage could be misleading as the mutation was induced in all of the transfected retinal progenitor cells, not only those in which this enhancer is active. The impact of the mutation (autonomous or non-cell autonomous) on the enhancer can distort the conclusion on the fate of the mutant cells, thereby complicating the interpretation of the relevance to Otx2 activity.

3) Referring to the control as "WT" is confusing. Note that the authors write that the "contralateral non-electroporated eye developed normally". This is not the control mentioned earlier. An accurate term should be provided for the relevant controls – those that were electroporated with Cas9 and guide scaffold. It is preferable to use a guide for an irrelevant target (LacZ for example) rather than a guide scaffold without a guide. Were the control eyes for the single-cell sequencing electroporated with Cas9 or not? This needs to be clarified.

---

## [Author Response]

Major comments:1) The mutation induced by Cas9 over time needs to be better characterized as providing information on the outcome of the Cas9 editing over time would be valuable data. You should evaluate the percentage of homozygous and heterozygous cells, preferably with respect to the unique indel types. Analyzing RNAseq data from populations of cells as shown in Figure 5—figure supplement 1 might be sufficient: The data show that most reads have indels, which could be useful for determining the percentage of homozygous and heterozygous cells and the indel types). This is important as the study is based on analyses of the outcome of a heterogeneous population of mutant cells that differ in the size and time of deletion.

We thank the reviewers for the suggestion to report additional details of the induced mutations. We have performed an additional experimental analysis to identify the mutations that were induced in OTX2 and provide quantitation of the frequencies of these mutations. We extracted mRNA from the cells left over upon completion of the 10X single cell sequencing experiment, generated cDNA and amplified an OTX2 region that encompassed the target region for the OTX2 guide 2. These PCR products were submitted for amplicon deep sequencing using a local sequencing provider (Genewiz). The analysis yielded a quantification of the sequences in both control and OTX2^CRISPR^ samples and revealed the types of mutation that occurred in the remaining OTX2 transcripts present in the single cell analysis upon OTX2 mutation. We updated the Materials and methods section, (subsection “Library preparation, sequencing and data analysis” paragraph four) as well as the Results, presented in paragraph four of subsection “Single cell RNA sequencing analysis of the OTX2ECR2-positive cells reveals widespread changes in the distribution of cells per cluster”, as well as Figure 5—figure supplement 1F, G. We did observe a higher percentage of wildtype reads than we expected given the penetrant phenotypic effects. This could be due to overrepresentation of the wildtype allele in our sequenced amplicons due to Cas9-induced deletion of one of the primer binding sites. In addition, it could be that mutant cells express a lower level of OTX2 as these cells change fate and therefore wildtype transcripts in the few remaining photoreceptors or other OTX2-positive cells are overrepresented at the RNA level. We note in the text that the number of OTX2-positive cells in the single cell datasets supports this possibility. It is also possible that OTX2 heterozygotes have haploinsufficient effects in this context and OTX2 has previously been shown to have such a genetic requirement in mice and humans. All of these possibilities are discussed in the Discussion, paragraph two of subsection “Timing and CRISPR/Cas9 effectiveness of the OTX2 ablation”.

Though this new analysis partially addresses the reviewers’ concern, we are unable to provide a specific estimation of the percentage of homozygous and heterozygous cells that are formed in response to the OTX2^CRISPR^ condition. We will first review why this is technically not feasible and then our response as to why we believe this analysis, though potentially informative, is not required for this study.

There are two ways in which reasonable estimates of the allele status of each single cell in the presented dataset could be determined. The first is by examination of the OTX2 mRNA species present in each cell. This would be the most direct method. However, due to the nature of the 10X RNASeq platform used in this experiment, the reads for all genes are heavily biased to the 3’ end of the gene and there is minimal representation of the Cas9 targeted region (Figure 5—figure supplement 1C) in OTX2, which is located near the 5’ region of the mRNA. The small number of reads that are present in this region all represent mutant alleles, which supports the hypothesis that the CRISPR/Cas9 system was in fact very efficient in this instance, and it is likely that many cells carry mutations in both alleles of OTX2.

The second method we could use is to extrapolate from the results of the bulk RT-PCR experiment and estimate homozygotes versus heterozygotes versus wildtype cell distributions. If we assumed that all of the wildtype alleles were present in the same cells (all cells were either wildtype or homozygous mutant with no heterozygotes present) we would expect that the ~29% wildtype alleles would represent 14.5% wildtype cells. At the other extreme, those wildtype alleles could always be paired with a mutant allele, thus all wildtype alleles would be in heterozygote cells. This would mean that 29% of cells were heterozygotes. However, it seems plausible that we could be underestimating the number of mutant alleles for the reasons noted above and this could lead to a misestimation of even this range of projected genotypes.

Though we agree with the reviewer that this would be a useful datapoint, we do not believe that it is required to support the findings of the current study. There have been several previous reports that OTX2 heterozygous mutations underlie human disease, so some of the observed phenotypes in the chicken retina may be in line with this sensitive genetic requirement for OTX2. Whether the results here are due to such a haploinsufficiency effect does not affect the findings that are the focus of this study. We have addressed this point in paragraph two of subsection “Timing and CRISPR/Cas9 effectiveness of the OTX2 ablation”.

2) You should also complete the analysis of loss of Otx2 at earlier stages.

We thank the reviewers for the suggestion and have responded by performing additional experiments to quantify the most encountered phenotypes that resemble clinical manifestation of people suffering OTX2 mutations, which are coloboma and microphthalmia. These results are described in paragraph two of subsection “CRISPR Cas9-induced ablation of the OTX2 gene at the optic vesicle stage results in anatomical defects of the overall eye structure, pigment epithelium and neuronal retina”, and detailed in Figure 1—figure supplement 1I, J and Supplementary file 2.

3) The use of reporter activity to monitor cell lineage could be misleading as the mutation was induced in all of the transfected retinal progenitor cells, and not only those in which this enhancer is active. The impact of the mutation (autonomous or cell non-autonomous) on the enhancer can distort the conclusion about the fate of the mutant cells, thereby complicating the interpretation of the relevance to Otx2 activity. You should discuss this limitation in the Discussion.

We appreciate the suggestion to explain the use of the reporter further as we believe it increases clarity of the study. We have included text in the Discussion subsection “Cell type-specific effects in response to loss of OTX2” paragraph two to discuss the possibility of cell non-autonomous mechanisms.

Other comments:– Referring to the control as "WT" is confusing. Note that you write that the "contralateral non-electroporated eye developed normally". This is not the control mentioned earlier. An accurate term should be provided for the relevant controls – those that were electroporated with Cas9 and guide scaffold. Were the control eyes for the single-cell sequencing electroporated with Cas9 or not? This needs to be clarified.

We replaced all abbreviation denoted initially as WT with the abbreviation for control – CTRL. We added a clearer explanation about the controls used: Results subsection “CRISPR Cas9-induced ablation of the OTX2 gene at the optic vesicle stage results in anatomical defects of the overall eye structure, pigment epithelium and neuronal retina” and “Single cell RNA sequencing analysis of the OTX2ECR2-positive cells reveals widespread changes in the distribution of cells per cluster”.

– Figure 1F: show a higher magnification of the mutant retina and RPE. Not clear if Otx2 is indeed reduced in the RPE.– Figure 1: also label the region encoding the homeodomain in Figure 1A. In the top panel (F-H), indicate the stages on the right (E1.5-E5), as was done for the lower panel.– Subsection “CRISPR Cas9-induced ablation of the OTX2 gene at the optic vesicle stage results in anatomical defects of the overall eye structure, pigment epithelium and neuronal retina”: revise the text to eliminate redundancy.

We thank the reviewers for the suggestions. We included a higher magnification image showing the area of the RPE that lack OTX2 expression, denoted by yellow arrowheads, in contrast with the few RPE cells that are OTX2-positive (white arrowheads).

We annotated Figure 1A, adding a light-grey bar representing the homeodomain.

We eliminated the redundancy in the text.

Reviewer #3:Comments:1) The mutation induced by Cas9 over time needs to be characterized with respect to the indel fraction, unique indel types, and percentage of homozygous and heterozygous cells. This is important as the study is based on analyses of the outcome of a heterogeneous population of mutant cells that differ in the size and time of deletion (see, for example, analyses presented in Figure 1 of the paper https://www.ncbi.nlm.nih.gov/pubmed/28985529).

We thank the reviewer for the suggestion. A similar analysis as in the above-mentioned paper is not possible, considering the differences in the experimental paradigms. The cited paper used single cells to seed colonies and then analyzed them. The experimental workflow our study used a scRNASeq analysis and the depth of sequencing, as well as the location of the target site relative to the sequencing reads, prevents such analysis. However, we did perform additional experiments using deep sequencing on the same sample that was analyzed at the single cell level. These data are described above in response to the Major Comments.

2) The use of reporter activity to monitor cell lineage could be misleading as the mutation was induced in all of the transfected retinal progenitor cells, not only those in which this enhancer is active. The impact of the mutation (autonomous or non-cell autonomous) on the enhancer can distort the conclusion on the fate of the mutant cells, thereby complicating the interpretation of the relevance to Otx2 activity.

We appreciate the suggestion to explain the use of the reporter further as we believe it increases clarity of the study. We addressed this matter in the Discussion subsection “Cell type-specific effects in response to loss of OTX2” paragraph two.

3) Referring to the control as "WT" is confusing. Note that the authors write that the "contralateral non-electroporated eye developed normally". This is not the control mentioned earlier. An accurate term should be provided for the relevant controls – those that were electroporated with Cas9 and guide scaffold. It is preferable to use a guide for an irrelevant target (LacZ for example) rather than a guide scaffold without a guide. Were the control eyes for the single-cell sequencing electroporated with Cas9 or not? This needs to be clarified.

We replaced all abbreviation denoted initially as WT with the abbreviation for control – CTRL. We added a clearer explanation about the controls used – see discussion above.